# Towards Dynamic Trend Filtering through Trend Points Detection with Reinforcement Learning

## Abstract

Trend filtering simplifies complex time series data by emphasizing proximity to the original data while applying smoothness to filter out noise. However, the inherent smoothness resulting from the 'approximateness' of trend filtering filters out the tail distribution of time series data, characterized as extreme values, thus failing to reflect abrupt changes in the trend. To address this, we draw inspiration from optimal stock trading strategy, which has to detect the lowest and highest points. As such, we reformulate the trend filtering problem by detecting essential points that should be reflected in the trend rather than approximations. In this paper, we introduce Trend Point Detection, a novel approach to identifying essential points to extract trends, which formulates the problem as a Markov Decision Process (MDP). We term these essential points as Dynamic Trend Points (DTPs) and extract trends by connecting them. To identify DTPs, we utilize Reinforcement Learning (RL) within a discrete action space, referred to as the Dynamic Trend Filtering network (DTF-net). DTF-net integrates flexible noise filtering, preserving critical original sub-sequences while removing noise as required for other sub-sequences. We demonstrate that DTF-net excels at capturing abrupt changes compared to other trend filtering algorithms, utilizing synthetic data and the Nasdaq intraday dataset. Furthermore, we demonstrate performance improvements in the forecasting task when we utilize DTF-net's trend as an additional feature, as abrupt changes are captured rather than smoothed out.

## 1 Introduction

Trend filtering emphasizes being close to the original time series data while filtering out noise through smoothness (Leser, 1961; Hodrick & Prescott, 1997; Kim et al., 2009). As such, trend filtering simplifies complex patterns within noisy and non-stationary time series data through smoothness, making it effective for tasks such as forecasting and anomaly detection (Rolski et al., 2009). In this context, smoothness eliminates noise and reveals the underlying structure of the time series pattern (Lin et al., 2017; Wen et al., 2019).

Traditional trend filtering employs a sum-of-squares loss function to achieve proximity to the original data while also utilizing second-order differences as a regularization term to attain smoothness (Hodrick & Prescott, 1997; Kim et al., 2009). However, we found that the constant nature of smoothness leads to filtering out not only noise but also abrupt changes that should be reflected in the trend. An 'abrupt change' denotes a point in a time series where the trend experiences a sharp transition. Given that the direction and persistence of the trend are determined by abrupt changes, it is vital to incorporate them into the trend.

The problem of constant smoothness stems from the property of 'approximateness' that trend filtering relies on, which has the effect of eliminating the tail distribution of the data. Ding et al. (2019) present evidence that sum-of-square loss approximates to Gaussian distribution, which has a light-tail shape and eliminates heavy-tail as outliers. Nevertheless, abrupt changes frequently reside within the tail distribution of time series data, making it challenging to distinguish them from the noise targeted for removal through smoothness (Kulik & Soulier, 2020). Therefore, constant smoothness applied in traditional trend filtering fails to distinguish abrupt changes and noise, result-

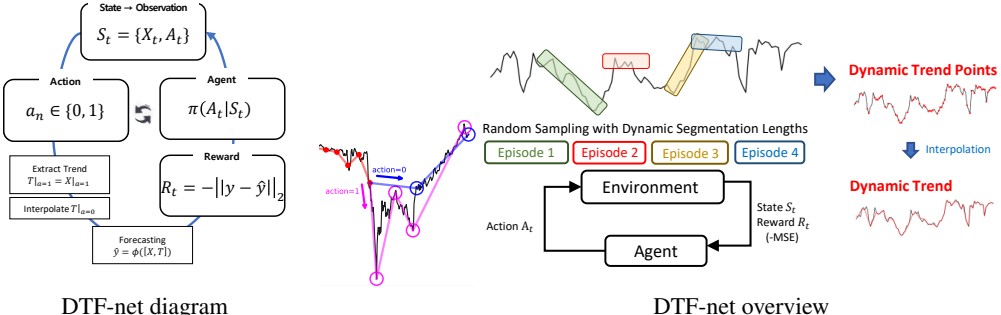

DTF-net diagram                    DTF-net overview

Figure 1: DTF-net extracts dynamic trends from time series data. Each episode is composed of a randomly sampled sub-sequence with a dynamic length from the entire time series data, and DTPs are determined based on action predictions. The final output consists of the detected DTP sequence, and the final trend is extracted through interpolation. The extraction of the trend varies depending on how the agent predicts the action.

ing in uniform filtering that leads to the loss of valuable information in trend. The heavier and longer the tail distribution, the problem of constant smoothness becomes aggravated (Wen et al., 2019).

Motivated by the nature of abrupt changes, categorized as extreme values, we introduce a novel algorithm designed to directly identify essential points for trend extraction, departing from approximations. This is inspired by the optimal stock trading strategy aimed at maximizing profit. Attaining the optimal strategy involves buying at the lowest point and selling at the highest point to achieve the maximum return, and these points align with the concept of abrupt changes and extreme values. In this context, Reinforcement Learning (RL) can capture abrupt changes, which are optimal trading points in a single stock trading task, by an agent. Derived from these motivations, we reformulate the Reinforcement Learning (RL) stock trading algorithm as a solution to the trend filtering problem.

In this paper, we introduce a novel approach to trend filtering that employs identifying essential points that should be reflected in the trend and subsequently interpolating them. These essential points are termed Dynamic Trend Points (DTPs), and the process of capturing them is referred to as Trend Point Detection. We formalize the Trend Point Detection problem as a Markov Decision Process (MDP) (Dynkin & Dynkin, 1965; Kaelbling et al., 1996; Sutton & Barto, 2018) and denote the algorithm that addresses it using RL (Schulman et al., 2017) as a Dynamic Trend Filtering network (DTF-net). In contrast to traditional trend filtering methods with approximateness, which have constant smoothness leading to filtering out abrupt changes, RL can directly detect these points through an agent. These essential points are dynamically detected by RL, unconstrained by fixed window sizes or frequencies within the time series data domain. This dynamic approach enables DTF-net to capture abrupt changes in the trend while adjusting the level of noise filtering for each sub-sequence within the time series (Sadhanala et al., 2017).

To achieve its purpose, DTF-net predicts discrete actions within a customized environment. We employ the Mean Squared Error (MSE) loss function from Time Series Forecasting (TSF) as a reward to capture temporal dependencies both before and after the action points. The degree of smoothness is controlled by the forecasting window size, which is a hyperparameter. Throughout DTF-net's training process, a bidirectional effect is achieved through iterative random sampling of segments from the entire time series data. Additionally, the overfitting issue can be mitigated by the random sampling method applied to the reward. To the best of our knowledge, this is the first approach that employs RL for trend filtering to simultaneously capture both abrupt changes and smoothness.

Our contributions are as follows:

- We found the issue of constant smoothness in traditional trend filtering, which is caused by the inherent property of 'approximateness', filters out both noise and abrupt changes. To address this problem, we directly detect points referred to as DTP that should be reflected in the trend.

- We formulate the problem of Trend Point Detection as MDP. Subsequently, we propose DTF-net, which employs RL, allowing it to filter trends while considering both smoothness and abrupt changes.

- We employ forecasting MSE cost function as a reward of DTF-net, allowing for the consideration of temporal dependencies when capturing DTPs. The sampling method is also applied to mitigate the overfitting issue.

- We demonstrate that DTF-net excels at capturing abrupt changes compared to other trend filtering methods and also enhances performance in forecasting tasks.

## 2 RELATED WORK

### 2.1 TREND FILTERING

Traditional trend-filtering algorithms attempted with various methods to capture abrupt changes. H-P (Hodrick & Prescott, 1997) and $\ell_1$ (Kim et al., 2009) stand out as widely used methods that optimize the sum of squared functions. However, they often suffer from delayed detection of abrupt changes due to the use of second-order difference operators. To tackle this issue, the TV-denoising algorithm (Chan et al., 2001) was introduced, relying on first-order differences. Nevertheless, this strategy introduces delays in detecting slow-varying trends. Additionally, due to the sum-of-squares loss, all the aforementioned trend filtering methods face challenges in handling heavy-tailed outliers (Wen et al., 2019).

Contrary to sum-of-squares loss methods, there are alternative approaches to trend filtering. First, frequency-dependent methods such as Wavelet (Rhif et al., 2019; Craigmile & Percival, 2002) are designed for non-stationary signals but are susceptible to overfitting. The Empirical Mode Decomposition (EMD) algorithm (Wu et al., 2007) employs a decomposition mechanism but can suffer from a mode-mixing effect. The common drawback shared by all aforementioned trend filtering algorithms is constant smoothness, resulting in undesired noise filtering even during abrupt changes.

### 2.2 EXTREME VALUE THEORY

Abrupt changes in a time series usually belong to the tail of the data distribution. The thicker and longer the tail, the greater the loss of information caused by the inherent constant smoothness of traditional trend filtering methods (Qian et al., 2020). In practice, the long heavy tail distribution diverges from the Gaussian distribution, which best represents a light-tailed bell shape. However, real-world time series data commonly exhibit long-heavy tail distributions as follows (von Bortkiewicz, 1921; Ding et al., 2019),

$$\lim_{T \to \infty} P\{max(y_1, ..., y_T) \leq y\} = \lim_{T \to \infty} F^T(y) = 0, \tag{1}$$

where $T$ random variables $y_1, ..., y_T$ are i.i.d. sampled from distribution $F_Y$. Furthermore, generalized extreme events can be modeled using the Extreme Value Theory.

**Theorem 1 (Extreme Value Theory (Fisher & Tippett, 1928; Ding et al., 2019))** *If the distribution in Equation 1 is not degenerate to 0 under the linear transformation on Y, the transformation of class with the non-degenerated distribution $G(y)$ should be the following distribution:*

$$G(y) = \begin{cases} exp(-(1 - \frac{1}{\gamma}y)^\gamma), & \gamma \neq 0, 1 - \frac{1}{\gamma}y \geq 0 \\ exp(-e^{-y}), & \gamma = 0 \end{cases} \tag{2}$$

Extreme Value Theory (EVT) revealed that extreme values exhibit a limited degree of freedom (Lorenz, 1963). This implies that the patterns of occurrence are recursive and can be memorized (Altmann & Kantz, 2005; Bunde et al., 2003). As a result, abrupt changes can be learned by a model with significant capacity and temporal invariance.

However, abrupt changes are either unlabeled or extremely imbalanced. Previous research (Fei-Fei et al., 2006; Vinyals et al., 2016; Wang et al., 2019) demonstrated the susceptibility of Deep Neural Networks (DNNs) to the data imbalance issue. Additionally, Ding et al. (2019) provided

evidence that minimizing Mean Squared Error loss presupposes a Gaussian distribution with variance $\tau$, expressed as $p(y_t|x_t, \theta) = \mathcal{N}(o_t, \tau^2)$, grounded in Bregman's theory, where given dataset $D = \{x \in X, y \in Y\}$ and model output $\{o \in O\}$ (Rosenblatt, 1956; Banerjee et al., 2005; Buch-Larsen et al., 2005; Singh & Gordon, 2008) (Appendix A). To solve this problem, they proposed Extreme Value Loss (EVL) to approximate extreme values in forecasting tasks, but it still fell short because of the inherent problem of 'approximateness'.

Therefore, our approach to the trend filtering problem is distinct from traditional methods that rely on approximating abrupt changes. Instead, we utilize RL, which enables the direct detection of abrupt changes. In other words, our primary objective shifts towards the challenge of training the agent to identify extreme values directly.

## 2.3 Markov Decision Process and Reinforcement Learning

Stock trading and portfolio optimization are among the most representative problems that can be modeled using time series data within an MDP (Liu et al., 2022a; Wu et al., 2020; Yang et al., 2020; Martinez et al., 2018). The decision-making process in stock trading, where actions depend solely on the immediate preceding state and are not influenced by the past, adheres to the Markov property. Consequently, it can be transformed into a problem of detecting trading points. Ideally, in stock trading, one should buy at the lowest turning point and sell at the highest turning point to maximize profit, and these turning points are equivalent to extreme value and abrupt changes. As motivated by this property, we define the problem of identifying essential points that should be reflected in the trend as an MDP, drawing parallels with this ideal trading strategy.

The Markov Process (MP) (Dynkin & Dynkin, 1965) serves as a model for representing the potential sequence of events in the future. This model is rooted in the Markov property, an assumption that exclusively influences the immediate event subsequent to the presently occurring one. MDP is built upon the first-order Markov assumption and comprises components denoted as $\langle S, A, P, R, \gamma \rangle$. In this context, $S$ denotes the set of environment states, while $A$ represents the set of actions undertaken by the agent at state $s$. The transition probability, $P = p(s\prime|s) = \Pr(S_{t+1} = s\prime|S_t = s)$, signifies the probability of transitioning from the current state $s$ to the next state $s\prime$. The reward, $R = r(s) = \mathbb{E}[R_{t+1}|S_t = s]$, originates from state $s$ when taking action $a \in A$. Finally, the discount factor $\gamma \in (0, 1]$ governs the trade-off between current and future rewards (Kaelbling et al., 1996; Sutton & Barto, 2018).

In MDP, actions are chosen through a policy network denoted as $\pi(a|s) = \Pr(A_t = a|S_t = s)$ for each state. Meanwhile, the state-value function $v_\pi(s)$ estimates the expected return value for a state $s$ under policy $\pi$. Approaches like A2C (Sutton & Barto, 2018) and PPO (Schulman et al., 2015; 2017) directly train agents using the Actor-Critic method. In this paradigm, the policy network $\pi$ is determined by Actor-Critic based on the estimated value function. In contrast, DQN (Mnih et al., 2013; 2015) defines the action-value function as $q_\pi(s, a)$ and aims to identify the optimal of $v^*$ and $q^*$ through the Bellman equation (Jaderberg et al., 2016; Silver et al., 2014) (Appendix E).

## 3 Dynamic Trend Filtering Network

### 3.1 Trend Point Detection and Markov Decision Process

#### 3.1.1 Environment Definition

DTF-net is a novel approach that employs RL to extract flexible trends from time series data. This approach effectively overcomes the limitations of existing trend-filtering algorithms, particularly with regard to capturing abrupt changes.

Time series data is defined as $\mathbf{T} = \{(\mathbf{X}_1, y_1), (\mathbf{X}_2, y_2), \ldots, (\mathbf{X}_N, y_N)\}$, where $\mathbf{X} \in \mathbb{R}^D$ represents the input, $y \in \mathbb{R}^d$ represents the output, and the dataset comprises a total of $N \in \mathbb{Z}^+$ samples. Here, $D$ and $d$ denote the respective dimensions of input and output.

We formalize Trend Point Detection as MDP.

- **State** $S = [\mathbf{X}_t, A_t]$: the positional encoded vector set of time series data $\mathbf{X}$ with horizon $t$ and detected action points $A \in \{0, 1\}$.

- **Action** $A$: a discrete action set with $a = 1$ for detecting DTP and $a = 0$ for smoothing.
- **Reward** $\mathcal{R}(S, A, S\prime)$: the change of the forecasting cost function value when action $A$ is taken at state $S$ and results in the transition to the next state $S\prime$.
- **Policy** $\pi(A|S)$: the probability distribution of $A$ at state $S$.

### 3.1.2   STATE AND ACTION FOR DTF-NET

Previous RL studies in time series typically followed a sequential approach. In contrast, DTF-net introduces dynamic segmentation with variable lengths within a single episode through random sampling based on discrete uniform distribution, given by

$$s \sim \text{unif}\{0, N\},$$

$$l \sim \text{unif}\{h + p, H\},$$

where $s$ denotes the starting points of the sub-sequence, $l$ denotes sub-sequence length, $h$ denotes horizon, $H$ denotes maximum horizon, and $p$ denotes forecasting horizon. This procedure dictates the sub-sequence's length and starting point, leading to a non-sequential and random progression of the episode. The overlap of sub-sequences as shown in Figure 1, allows us to achieve the advantages of bidirectional learning.

However, the varying length of the state with different episodes poses a problem. To tackle this issue, we maintain a constant state length through positional encoding.

$$PE_{(pos,2i)} = sin(pos/10000^{2i/d_{model}}),$$

$$PE_{(pos,2i+1)} = cos(pos/10000^{2i/d_{model}}).$$

Additionally, within a single episode, DTF-net differs from previous research by constructing the state cumulatively, rather than using a fixed window horizon. In other words, with each step within the episode, the length of the state gradually increases as follows,

$$E = \mathbf{T}_{s:s+l},$$

$$S_0 = PE(E_0),$$

$$S_{t+1} = PE(E_{0:t}, A_{0:t}).$$

Consequently, the state consists of both time series data $\mathbf{X}$ and action $A$, and through this cumulative state, the agent can learn sequential information from the time series (Appendix C.1).

As for the action space, DTF-net aims to identify important points that should be included in the trend. Therefore, the action space is defined as a discrete set of $\{0, 1\}$, where the agent predicts whether a particular time step is an essential trend point or not.

### 3.2   REWARD OF DTF-NET

In the stock trading task, RL is optimized based on cumulative return as a reward. However, in general time series, we cannot specify return as a reward. As a result, similar to traditional trend filtering, DTF-net aims to preserve the 'closeness to the original data' property, fundamental to trend filtering. However, employing a sum of squared loss function on current values makes it challenging to reflect appropriate smoothness. Therefore, DTF-net distinguishes itself by optimizing for future values instead of current ones. By employing Time Series Forecasting (TSF) and training RL to minimize its cost function, DTF-net can reflect the characteristics of each sub-sequence and learn temporal dependencies. This allows adjusting smoothness by tuning the prediction window.

As shown in Algorithm 1, the reward process involves time series data $\mathbf{X}$ and action $A$ in episode $E_{t-(h+p):t}$ at time step $t$, both having a sequence length denoted as $h + p$, where $h$ denotes the past horizon, and $p$ denotes the forecasting horizon. The trend $\mathcal{T}$ initiates with null values and values are assigned only under the condition of action $a = 1$. For the remaining null values, linear interpolation is applied. Subsequently, forecasting is conducted with a prediction length $p$ defined by a hyperparameter. The reward is computed as the negative MSE loss between the predicted $\hat{y}$ and the ground truth $y$. As illustrated in Figure 2, effective capture of abrupt changes leads to improved

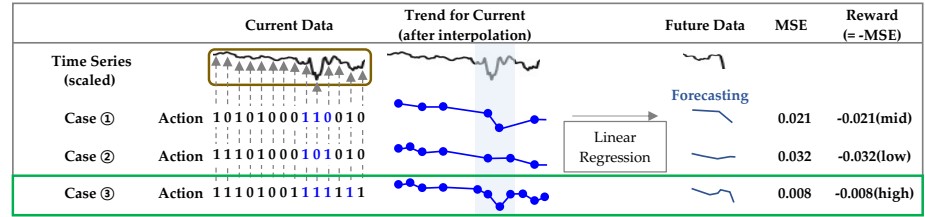

Figure 2: How does the reward optimize DTF-net to extract dynamic trends that incorporate abrupt changes? As shown in case 3, when DTF-net successfully identifies abrupt changes (blue action points), the prediction outcomes improve significantly, resulting in the highest reward.

---

**Algorithm 1** DTF-net Reward

1: **procedure** REWARD($E_{t-(h+p):t}$)          ▷ $\mathbf{X}, A \in E_{t-(h+p):t}$ at time step $t$
2:      $\mathcal{T} \leftarrow 0$          ▷ trend initialization
3:      $\mathcal{T}|_{a=1} \leftarrow \mathbf{X}|_{a=1}$          ▷ value assign for a=1
4:      **while** $n \leq h$ **do**          ▷ for linear interpolation
5:          // $n$ for time-axis and $\mathbf{x} \in \mathbf{X}$
6:          $\mathcal{T}_n \leftarrow \mathbf{x}_n = \mathbf{x}_{n-1} + \frac{\mathbf{x}_{n+1} - \mathbf{x}_{n-1}}{2}$
7:          $n \leftarrow n + 1$
8:      **end while**
9:      $\hat{y} \leftarrow \phi([\mathbf{X}, \mathcal{T}])$          ▷ Regression for TSF
10:      $r \leftarrow \frac{1}{p} \sum_{i=1}^{p} (y_i - \hat{y}_i)^2$          ▷ MSE loss
11:      **return** $-r$          ▷ minus of forecasting reward
12: **end procedure**

---

forecasting performance and a higher final reward. Additionally, to mitigate overfitting issues, DTF-net utilizes random sampling from a discrete uniform distribution of calculating reward at every time step, providing better control over model updates as follows,

$$k \sim \text{unif}\{s, s + l\},$$

$$R = \begin{cases} \text{REWARD}(E_{t-(h+p):t}) & \text{if } t = k, \\ 0 & \text{if } t \neq k. \end{cases}$$

In here, we found that by using a penalty reward as a negative value, irregularly applying penalties through sampling can prevent overfitting rather than penalizing at every moment (Appendix C.2). Note that the model $\phi$ employed within the environment can be any basic machine learning model, such as Linear Regression.

We use PPO (Schulman et al., 2017) with MLP-policy to learn discrete action (Appendix E).

## 4 EXPERIMENT

### 4.1 TREND FILTERING ANALYSIS ON SYNTHETIC DATA

#### 4.1.1 EXPERIMENTAL SETTING

Analyzing trend filtering methods poses two challenges. First, defining a ground truth for the trend is challenging. For example, in the case of $\ell_1$, the degree of smoothness varies with $\lambda$, resulting in different trends. Second, labeling abrupt changes is challenging. Obtaining real-world data with such labels is difficult, and arbitrary assignment to real-world data is subjective. To address this, we generate a synthetic trend signal with $1,000$ data points, featuring sine and triangle waves with varying amplitudes. The synthetic dataset has 11 abrupt changes, including gradual variations from data points 121 to 250, abrupt changes between data points 500 and 719, a sudden drop at data point

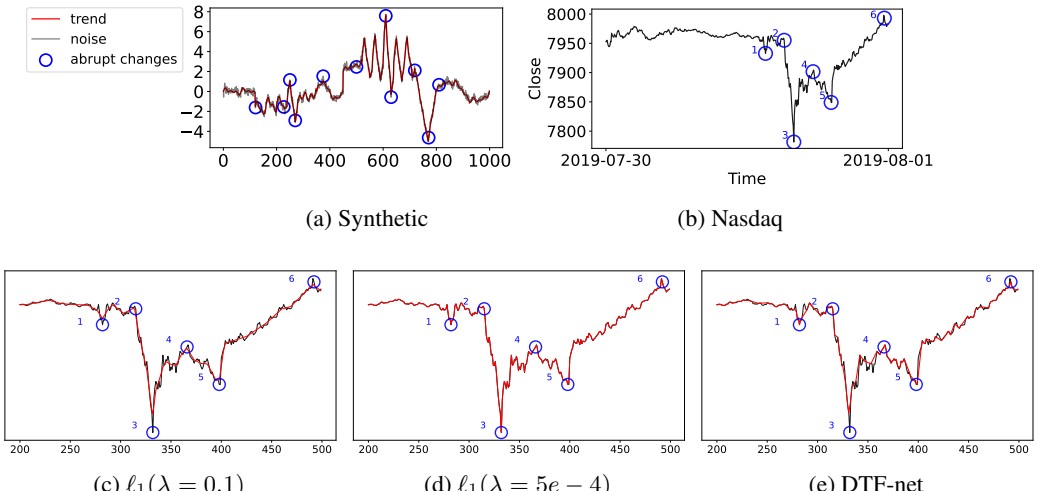

Figure 3: Figures (a) and (b) show the data that contains abrupt change with mean and variance shifts. In contrast to the underfitting or overfitting of trends caused by the constant smoothness of $\ell_1$ trend filtering, (c) and (d), DTF-net applies different levels of smoothness to each sub-sequence, enabling it to capture abrupt changes while dynamically performing noise filtering (e).

| Trend Filtering | Linear Signal | | | | Linear Signal+Noise (0.2) | | | |
| --- | --- | --- | --- | --- | --- | --- | --- | --- |
| | 1) full-sequence | | 2) abrupt-sequence | | 1) full-sequence | | 2) abrupt-sequence | |
| | MSE | MAE | MSE | MAE | MSE | MAE | MSE | MAE |
| ADAGA (Wu et al., 2020) | 4.3507 | 1.4319 | 7.0179 | 1.8476 | 4.3434 | 1.4428 | 7.0120 | 1.8668 |
| RED-SDS (Ansari et al., 2021) | 0.9678 | 0.6566 | 1.6329 | 0.9169 | 1.0036 | 0.6782 | 1.6660 | 0.9365 |
| TimesNet(Wu et al., 2023) | 3.0047 | 1.4112 | 3.2740 | 1.4161 | 3.0841 | 1.4204 | 3.3304 | 1.4364 |
| EMD (Wu et al., 2007) | 5.2836 | 1.7294 | 6.4599 | 1.8313 | 5.3096 | 1.7401 | 6.4410 | 1.8431 |
| Median (Siegel, 1982) | 4.4506 | 1.5335 | 5.7018 | 1.8099 | 4.4766 | 1.5525 | 5.6859 | 1.8204 |
| H-P (Hodrick & Prescott, 1997) | 0.1881 | 0.2807 | 0.2923 | 0.3493 | 0.2253 | 0.3311 | 0.3238 | 0.3934 |
| Wavelet (Rhif et al., 2019) | 0.0427 | 0.1676 | 0.0451 | 0.1740 | $1e-30$ | $6e-16$ | $2e-30$ | $8e-16$ |
| $\ell_1$ ($\lambda$=0.1) (Kim et al., 2009) | **0.0150** | **0.0885** | **0.0166** | **0.0971** | 0.0461 | 0.1703 | 0.0500 | 0.1807 |
| $\ell_1$ ($\lambda = 5e-4$) | 0.0379 | 0.1570 | 0.0403 | 0.1638 | 0.0004 | 0.0175 | 0.0004 | 0.0174 |
| DTF-net (ours) | 0.0378 | 0.1554 | 0.0389 | 0.1608 | **0.0289** | **0.0826** | **0.0286** | **0.0855** |

Table 1: We conduct trend filtering analysis on synthetic data, evaluating it with two ground truths: only the linear signal and the linear signal with added noise. We consider two cases, one with the full sequence and the other with a sub-sequence containing abrupt changes. The evaluation metrics include MSE and MAE, where lower values indicate better performance. The best performance is **bolded**, and the second-best performance is underlined.

771, and the initiation of a sine wave at data point 810. We add Gaussian noise with a standard deviation of 0.2 to simulate real-world conditions.

To evaluate the trend filtering results, we set the experiment as follows. First, we use generated abrupt changes as oracle points to evaluate abrupt change capture. Performance evaluations include a 30-window interval around abrupt changes to assess temporal dependencies effectively. Second, we employ Mean Squared Error (MSE) and Mean Absolute Error (MAE) metrics to measure proximity to the original data, a key aspect of trend filtering. For ground truth, we use a linear signal and a linear signal with added noise to assess robustness to noisy data. In cases with added noise, we assume that filtering out at least 10% of noise is necessary to confirm smoothness. Regarding the comparative approaches, we compare with five prominent trend filtering algorithms, recent CPD algorithms, and recent anomaly detection algorithms (Appendix F)

### 4.1.2 PERFORMANCE ANALYSIS

**Synthetic Dataset** As shown in Table 1, when considering the ground truth of a linear signal, DTF-net outperforms other prominent trend filtering methods, CPD, and anomaly detection algorithms,

| Methods | | DTF-Linear (ours) | | $\ell_1(\lambda=0.1)$-Linear | | PatchTST/42 | | NLinear | | DLinear | | FEDformer-f | | FEDformer-w | | Autoformer | |
|---|---|---|---|---|---|---|---|---|---|---|---|---|---|---|---|---|---|
| Metric | | MSE | MAE | MSE | MAE | MSE | MAE | MSE | MAE | MSE | MAE | MSE | MAE | MSE | MAE | MSE | MAE |
| Exchange | 24 | **0.0250** | **0.1198** | 0.0266 | 0.1248 | 0.0387 | 0.1513 | 0.0275 | 0.1264 | 0.0290 | 0.1284 | 0.0381 | 0.1545 | 0.0387 | 0.1564 | 0.0687 | 0.2041 |
| | 48 | **0.0487** | **0.1658** | 0.0505 | 0.1708 | 0.0624 | 0.1873 | 0.0505 | 0.1705 | 0.0585 | 0.1907 | 0.0548 | 0.1818 | 0.1068 | 0.2528 | 0.1095 | 0.2485 |
| | 96 | **0.098** | **0.2349** | 0.1007 | 0.2440 | 0.1833 | 0.3436 | 0.0990 | 0.2361 | 0.1063 | 0.2530 | 0.1440 | 0.2980 | 0.1386 | 0.2894 | 0.1834 | 0.3306 |
| | 192 | 0.1983 | 0.3583 | 0.2045 | 0.3518 | 0.2550 | 0.3987 | 0.2030 | 0.3400 | 0.1959 | 0.3554 | 0.2790 | 0.4163 | 0.2841 | 0.4217 | 0.3465 | 0.4510 |
| | 336 | **0.3160** | **0.4561** | 0.3337 | 0.4666 | 0.5161 | 0.5442 | 0.4174 | 0.4857 | 0.3276 | 0.4627 | 0.4466 | 0.5130 | 0.5685 | 0.5890 | 0.4488 | 0.5291 |
| | 720 | **0.7933** | **0.6874** | 0.9515 | 0.7636 | 1.1143 | 0.8063 | 1.0420 | 0.7807 | 0.9071 | 0.7415 | 1.2122 | 0.8492 | 1.2912 | 0.8876 | 1.2463 | 0.8694 |
| ETTh1 | 24 | 0.0253 | 0.1205 | 0.0234 | 0.1140 | 0.0266 | 0.1238 | 0.0266 | 0.1240 | 0.0273 | 0.1262 | 0.0358 | 0.1450 | 0.0381 | 0.1524 | 0.0694 | 0.2042 |
| | 48 | 0.0375 | 0.1479 | 0.0366 | 0.1442 | 0.0393 | 0.1506 | 0.0393 | 0.1503 | 0.0404 | 0.1523 | 0.0547 | 0.1778 | 0.0602 | 0.1921 | 0.0797 | 0.2205 |
| | 96 | **0.0519** | **0.1740** | 0.0521 | 0.1744 | 0.0550 | 0.1790 | 0.0519 | 0.1745 | 0.0551 | 0.1815 | 0.0786 | 0.2126 | 0.0919 | 0.2348 | 0.0857 | 0.2292 |
| | 192 | **0.0676** | **0.2013** | 0.0693 | 0.2034 | 0.0705 | 0.2050 | 0.0694 | 0.2046 | 0.0730 | 0.2076 | 0.0933 | 0.2344 | 0.1000 | 0.2464 | 0.0993 | 0.2428 |
| | 336 | 0.0803 | 0.2247 | 0.0796 | 0.2238 | 0.0814 | 0.2260 | 0.0826 | 0.2280 | 0.0948 | 0.2414 | 0.1117 | 0.2597 | 0.1418 | 0.2958 | 0.1287 | 0.2792 |
| | 720 | **0.0776** | **0.2224** | 0.0789 | 0.2244 | 0.0869 | 0.2329 | 0.0814 | 0.2273 | 0.1800 | 0.3494 | 0.1310 | 0.2858 | 0.1224 | 0.2766 | 0.1378 | 0.2939 |
| Illness | 24 | **0.5881** | **0.5358** | 0.6119 | 0.5299 | 0.6228 | 0.5305 | 0.6325 | 0.5639 | 0.7831 | 0.7462 | 0.6969 | 0.6256 | 0.7100 | 0.6352 | 0.7432 | 0.6704 |
| | 48 | **0.6858** | **0.6359** | 0.6925 | 0.6322 | 0.7109 | 0.6642 | 0.6892 | 0.6453 | 0.8217 | 0.7750 | 0.7099 | 0.6935 | 0.6961 | 0.6972 | 0.7855 | 0.7370 |
| | 60 | 0.6640 | 0.6423 | 0.6666 | 0.6324 | 0.6465 | 0.6381 | 0.6730 | 0.6347 | 0.9195 | 0.8361 | 0.8309 | 0.7653 | 0.8192 | 0.7641 | 0.8945 | 0.8055 |

Table 2: We conduct TSF experiments using three datasets: Exchange Rate, ETTh1, and Illness. We evaluate performance using MSE and MAE, where lower values indicate better performance. In the following results, the best-performing models using DTF-net are highlighted in **bold**, and models using $\ell_1$ trend filtering are highlighted in *italic*. To compare the results, the best-performing models using the original data are underlined.

except for $\ell_1(\lambda = 0.1)$. This is attributed to the piece-wise linearity assumption of the $\ell_1$ method, extracting more linear features than DTF-net, which prioritizes abrupt change consideration over linearity. However, our goal is to reflect abrupt changes within noise. Therefore, with the ground truth of a linear signal added with noise, DTF-net surpasses all prominent methods, except for those prone to overfitting. As trend filtering should also consider smoothness, we assume a minimum degree of noise filtering, set at 10% of Gaussian noise. Consequently, Wavelet and $\ell_1(\lambda = 5e - 4)$ can be considered as overfitting to noisy data. These results suggest that DTF-net exhibits an enhanced ability to dynamically capture abrupt changes within noisy and complex time series data.

**Real-World Dataset** To demonstrate the proficiency of DTF-net on complex datasets, we perform additional trend-filtering analysis using the Nasdaq intraday dataset from July 30th to August 1st, 2019, characterized by rapid changes. Here, we arbitrarily set 6 abrupt changes and qualitatively analyze the results. As shown in Figure 3, it is evident that the $\ell_1$ trend filtering algorithm extracts trends that either underfit or overfit depending on the parameter $\lambda$ due to constant smoothness. In contrast, DTF-net accurately captures all six data points and concurrently performs noise filtering for point 3. This accomplishment is attributed to the dynamic nature of trend extraction within DTF-net.

## 4.2 Trend Filtering Analysis on Real-World Datasets

### 4.2.1 Experimental Settings

Defining the ground truth for trend filtering poses challenges; hence, we indirectly compare methods within the application to Time Series Forecasting (TSF). TSF models are expected to signal potential incidents related to extreme values, aiding in critical decision-making (Okubo & Narita, 1980; Van den Berg et al., 2008). Therefore, to evaluate the practicality of DTF-net in real-world scenarios, we apply it to TSF, where we incorporate the extracted trend as an additional input feature. Formally, the forecasting model receives input as $\mathbf{X}\prime = [\mathbf{X}, \mathbf{P}] \in \mathbb{R}^{D+d}$, with $\mathbf{P}$ representing the trend from DTF-net. Under the same conditions, we compare this model to those using $\ell_1$ as additional inputs and only the original sequence $\mathbf{X}$ as inputs. We employ the TSF model with NLinear and DLinear (Zeng et al., 2022), both considered state-of-the-art but simplest in TSF. The experiment focuses on the univariate case to assess trend filtering effectiveness (Appendix B).

### 4.2.2 Performance Analysis

We specifically select three non-stationary datasets from the TSF benchmark dataset: Exchange Rate, ETTh1, and Illness. Table 2 indicates that DTF-net outperforms in most cases. In detail, among the three datasets, the exchange rate dataset is the most intricate, which exhibits the least seasonality and the highest level of noise. Given the absence of periodicity in financial data, $\ell_1$ trend filtering encounters difficulties in extracting clear trends. However, DTF-net demonstrates robustness when dealing with non-stationary time series data.

However, models employing $\ell_1$ trend filtering, which assumes piece-wise linearity, offer advantages when dealing with data that exhibits a previous step motion. This linearity is particularly pronounced

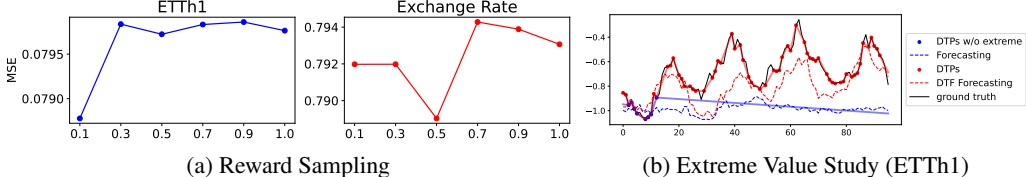

| (a) Reward Sampling | (b) Extreme Value Study (ETTh1) |

Figure 4: 1) The figure shows how DTF-net addresses overfitting. The $x$-axis denotes the reward sampling ratio, while the $y$-axis represents the MSE. A downward trend indicates better performance. Each dataset has an optimal reward ratio, and beyond that point, overfitting occurs. 2) We conduct experiments to evaluate the influence of trends that incorporate extreme values with long-heavy tails on forecasting. The figure shows that including extreme values (red) in forecasting plays a crucial role without undergoing smoothing (blue).

in short-term predictions within ETTh1, as it is the least noisy and most stationary dataset among the three. As shown in Table 2, $\ell_1$ achieves the best results for 24- and 48-hour forecasting windows in ETTh1. While DTF-net also outperforms the single forecasting model, the linearity characteristic of $\ell_1$ is better suited for short-term predictions within ETTh1. In contrast, for long-term predictions, we demonstrate that DTF performs the best. In the case of the Illness with a small dataset size, DTF-net also performs well without overfitting.

### 4.2.3 Ablation Study

**How to handle overfitting of RL?** To mitigate the risk of overfitting in RL-based trend filtering, we introduce a reward sampling method. This approach offers better control over model updates. As shown in Figure 4-(a) for the ETTh1 dataset (in blue), reward sampling effectively prevents overfitting, achieving optimal performance with a reward ratio of 0.1. Similarly, for the Exchange Rate dataset (in red), optimal performance is observed with a reward ratio of 0.5. Increasing the reward ratio leads to overfitting and a subsequent decline in performance. Therefore, adjusting the reward sampling ratio proves effective in mitigating the overfitting problem.

**Empirical analysis on extreme value** While DNNs typically enhance MSE performance through empirical risk minimization, they tend to produce smooth, averaged predictions in regression tasks. However, our objective is to incorporate vital abrupt changes into trends, resulting in predictions that accurately represent both upward and downward trends rather than providing solely smooth estimates. Qualitatively, we aim to demonstrate that our methodology better tracks crucial peaks compared to trends that do not consider extreme values in the prediction task.

We evaluate the impact of trends with extreme values on forecasting tasks by comparing two trends. The red line includes all DTPs, while the blue line excludes 10% of extreme values. In Figure 4-(b) for the ETTh1 dataset (pred=24), the blue line struggles to capture abrupt changes effectively, especially in segments with a variance shift. Consequently, predictions based on the blue dashed line tend to appear overly smooth. In contrast, forecasts from the trend incorporating extreme values (illustrated by the red dashed line) exhibit enhanced performance, capturing peaks in the prediction output and resulting in more accurate forecasts.

## 5 Conclusion

We propose DTF-net, a novel trend filtering approach that directly captures abrupt changes using RL. Traditional methods fail to capture abrupt changes due to constant smoothness based on approximateness property, whereas DTF-net, directly identifies these points. We define DTPs as essential trend points and formulate a Trend Point Detection problem as MDP. Using a discrete action space, the agent identifies critical points, with the reward defined as the MSE of forecasting. In each RL episode, randomly sampled segments with varying lengths overlap, facilitating bidirectional learning. The MSE reward allows DTF-net to capture temporal dependencies while extracting trends. We demonstrate that DTF-net outperforms in identifying abrupt changes using synthetic and Nasdaq intraday datasets. Additionally, DTF-net excels in predicting by considering crucial peak points in real-world TSF tasks.

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

# A   EXTENDED PROOFS FOR APPROXIMATENESS AND GAUSSIAN DISTRIBUTION

Ding et al. (2019) provided evidence that minimizing Equation 3 assumes a Gaussian distribution with variance $\tau$, such that $p(y_t|x_t, \theta) = \mathcal{N}(o_t, \tau^2)$, based on Bregman's theory Banerjee et al. (2005); Singh & Gordon (2008).

$$
\begin{aligned}
\hat{P}(Y) &= \min \sum_{t=1}^{T} ||o_t - y_t||^2, \\
&= \max_\theta \Pi_{t=1}^T P(y_t|x_t, \theta), \\
&= \frac{1}{N} \sum_{t=1}^{T} \mathcal{N}(y_t, \hat{\tau}^2),
\end{aligned}
\tag{3}
$$

under the given considerations:

- There is no prior distribution on the discriminative model, and
- The output $o_t$ is learned assuming a likelihood such as normal distribution,

the resulting form be similar to a Kernel Density Estimator (KDE) with a Gaussian kernel (Rosenblatt, 1956). This indirectly demonstrates that DNNs with a traditional loss function may not perform well on extreme values present in heavy-tailed distributions (Buch-Larsen et al., 2005; Ding et al., 2019).

# B   EXTENDED EXPERIMENTS ON TSF

## B.1   TSF DATASET

We specially choose chaotic datasets in the TSF benchmark for evaluating DTF-net is well-capturing abrupt changes (Liu et al., 2022b).

| Dataset | Variable Number | Sampling Frequency | ADF Test Statistic |
|---|---|---|---|
| Exchange | 8 | 1Day | -1.889 |
| ILI | 7 | 1Week | -5.406 |
| ETT | 7 | 1Hour / 15Minutes | -6.225 |
| Electricity | 321 | 1Hour | -8.483 |
| Traffic | 862 | 1Hour | -15.046 |
| Weather | 21 | 10Minutes | -26.661 |

Table 3: Summary of TSF datasets. Smaller ADF test statistic indicates a more stationary dataset.

- `ETTh1`: Electricity Transformer Temperature for a 1-hour period. The data is gathered over a two-year period from two different Chinese countries. The target value "Oil Temperature" and six power load features make up each data point (Zhou et al., 2021).
- `Exchange Rate`: The collection of the daily exchange rates of eight foreign countries including Australia, British, Canada, Switzerland, China, Japan, New Zealand, and Singapore ranging from 1990 to 2016 (Lai et al., 2018).
- `Illness`: Patient data recorded for influenza illness weekly from the US Centers for Disease Control and Prevention between 2002 and 2021. This dataset shows the ratio of patients seen with influenzalike illness and the number of patients (Wu et al., 2021).

## B.2   TSF BASELINES.

- `Autoformer`: As a Transformer-based method, Autoformer learns the temporal pattern of time series by decomposition and Auto-Correlation mechanism through Fast Fourier Transform (Wu et al., 2021).

- `FEDformer`: As a Transformer-based method, FEDformer introduced a Mixture of Experts (MOE) for seasonal-trend decomposition and frequency-enhanced block/attention with Fourier and Wavelet Transform (Zhou et al., 2022).

- `DLinear`: Only using Linear layers, DLinear decomposes the original input into a trend and remainder components. Then, two linear layers are applied to each component and sum up the two features to obtain the final prediction (Zeng et al., 2022).

- `NLinear`: To overcome the train-test distribution shift in the dataset, NLinear uses a simple normalization that subtracts the last value from the input and adds it back before making the final prediction (Zeng et al., 2022).

- `PatchTST`: As a transformer-based model, PatchTST has two components: segmentation of time series into subseries-level patches, and channel-independence structure. PatchTST can capture local semantic information and benefit from longer look-back windows (Nie et al., 2022).

| Dataset | Input Length | Prediction Length | Forecasting Model | Reward Ratio | Learning Rate | RL Epoch | Forecasting Epoch | Max Sequence Length |
|---|---|---|---|---|---|---|---|---|
| | 336 | 720 | DLinear | 0.4 | 5e-4 | 10000 | 15 | 3000 |
| | 336 | 336 | DLinear | 0.4 | 5e-4 | 10000 | 15 | 3000 |
| Exchange Rate | 336 | 192 | DLinear | 0.4 | 1e-4 | 10000 | 15 | 3000 |
| | 336 | 96 | NLinear | 0.4 | 1e-4 | 3000 | 15 | 3000 |
| | 336 | 48 | NLinear | 0.4 | 1e-4 | 3000 | 15 | 3000 |
| | 336 | 24 | NLinear | 0.4 | 1e-4 | 3000 | 15 | 3000 |
| | 336 | 720 | NLinear | 0.1 | 1e-3 | 10000 | 20 | 3000 |
| | 336 | 336 | NLinear | 0.1 | 9e-4 | 10000 | 15 | 3000 |
| ETTh1 | 336 | 192 | NLinear | 0.1 | 3e-4 | 10000 | 15 | 3000 |
| | 336 | 96 | NLinear | 0.1 | 5e-4 | 1000 | 15 | 3000 |
| | 336 | 48 | NLinear | 0.1 | 5e-4 | 1000 | 15 | 3000 |
| | 336 | 24 | NLinear | 0.1 | 5e-4 | 1000 | 15 | 3000 |
| | 104 | 60 | NLinear | 0.1 | 1e-3 | 1000 | 15 | 300 |
| Illness | 104 | 48 | NLinear | 0.1 | 1e-3 | 5000 | 15 | 300 |
| | 104 | 24 | NLinear | 0.1 | 1e-3 | 10000 | 15 | 300 |

Table 4: Hyper-Parameters.

As for baseline, the learning rate follows Table 4, but when overfitting occurs it downgrades to $1e-4$ according to (Zeng et al., 2022). Other conditions are the same as follows: Adam optimizer, MSE loss function, 15 epochs, and 32 batch sizes. Note that Linear models use 336 as the input sequence length, and the Transformer model uses 96. DTF-net uses PPO (Schulman et al., 2017) to extract trends. The seed number was arbitrarily set to 2023.

## B.3 EXTENDED EXTREME VALUE STUDY ON TSF

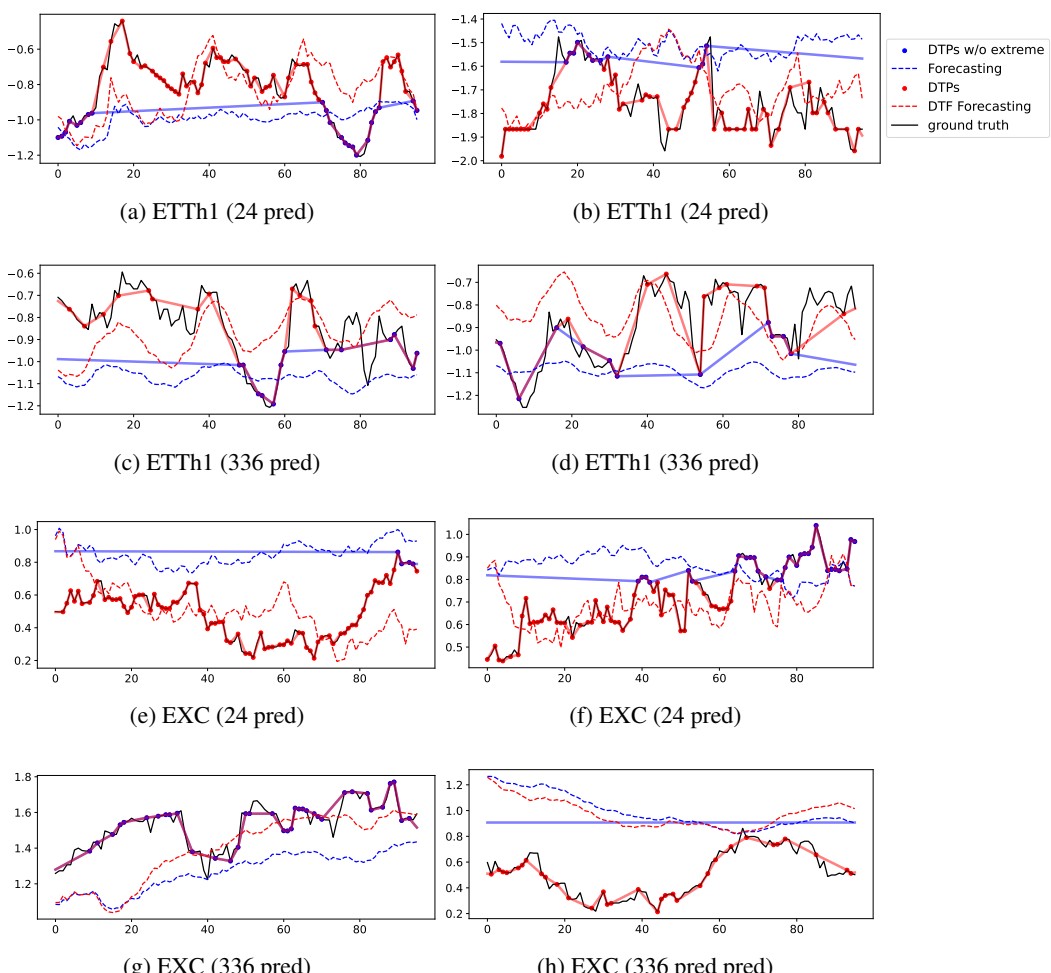

Figure 5: In addition to Figure 4-(b), we provide a qualitative example to demonstrate the importance of including extreme values in the trend. Through two datasets, ETTh1 and Exchange rate (EXC), we can observe that when extreme values are removed (indicated by the blue line) in both short-term (24 pred) and long-term (336 pred) forecasting, the forecasting results also appear to be smoother.

## C EXTENDED ABLATION STUDY ON DTF-NET

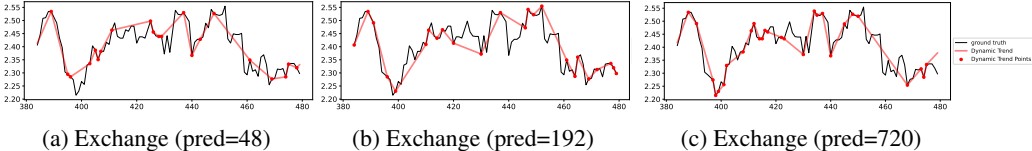

| (a) Exchange (pred=48) | (b) Exchange (pred=192) | (c) Exchange (pred=720) |

Figure 6: The figure shows how trends extracted by DTF-net vary within the same sub-sequence of the exchange rate dataset based on different prediction horizon sizes. As the prediction horizon increases, Dynamic Trend Points (DTPs) are captured in more detail, as exemplified by the index at 400. Compared to existing trend filtering methods, DTF-net indicates its ability to extract more dynamic trends.

### C.1 ABLATION STUDY ON STATE

| State | Episode length | non-sequential dynamic | | non-sequential static | | sequential dynamic | | sequential static | | zero padding | |
|---|---|---|---|---|---|---|---|---|---|---|---|
| Metric | | MSE | MAE | MSE | MAE | MSE | MAE | MSE | MAE | MSE | MAE |
| Exchange | 24 | **0.0250** | **0.1198** | 0.0263 | 0.1228 | 0.0264 | 0.1231 | 0.0263 | 0.1227 | 0.0259 | 0.1215 |
| | 48 | **0.0487** | **0.1658** | 0.0500 | 0.1697 | 0.0496 | 0.1689 | 0.0501 | 0.1699 | **0.0486** | **0.1655** |
| | 96 | 0.0983 | 0.2349 | 0.0995 | 0.2363 | 0.0994 | 0.2363 | **0.0982** | **0.2348** | 0.0983 | 0.2350 |
| | 192 | **0.1983** | **0.3583** | 0.1986 | 0.3587 | 0.2013 | 0.3607 | 0.1984 | 0.3582 | 0.2003 | 0.3598 |
| | 336 | 0.3160 | 0.4561 | 0.3163 | 0.4562 | 0.3166 | 0.4562 | **0.3140** | **0.4541** | 0.3166 | 0.4564 |
| | 720 | 0.7933 | 0.6874 | 0.7922 | **0.6822** | 0.7903 | 0.6878 | 0.7893 | 0.6874 | **0.7889** | **0.6871** |
| ETTh1 | 24 | **0.0253** | **0.1205** | 0.0264 | 0.1228 | 0.0255 | 0.1205 | 0.0259 | 0.1214 | 0.0258 | 0.1216 |
| | 48 | **0.0375** | 0.1479 | 0.0381 | 0.1487 | 0.0381 | 0.1485 | 0.0379 | **0.1478** | 0.0379 | 0.1479 |
| | 96 | **0.0519** | **0.1740** | 0.0551 | 0.1810 | 0.0554 | 0.1808 | 0.0553 | 0.1808 | 0.0555 | 0.1811 |
| | 192 | **0.0676** | **0.2013** | 0.0680 | 0.2019 | 0.0700 | 0.2051 | 0.0687 | 0.2026 | 0.0695 | 0.2041 |
| | 336 | 0.0803 | 0.2247 | 0.0805 | 0.2252 | **0.0796** | **0.2238** | 0.0806 | 0.2254 | 0.0803 | **0.2244** |
| | 720 | **0.0776** | **0.2224** | 0.0808 | 0.2271 | 0.0809 | 0.2273 | 0.0808 | 0.2273 | 0.0795 | 0.2255 |
| Illness | 24 | 0.5881 | 0.5358 | 0.5805 | 0.5363 | 0.5845 | 0.5376 | **0.5621** | **0.5316** | **0.5808** | 0.5464 |
| | 48 | 0.6858 | 0.6359 | 0.6558 | 0.6329 | 0.6813 | 0.6535 | **0.6255** | **0.5964** | **0.6551** | **0.6310** |
| | 60 | 0.6640 | 0.6423 | 0.7481 | 0.7029 | **0.6506** | **0.6265** | 0.7455 | 0.6979 | **0.6513** | **0.6270** |

Table 5: **State ablation study results.** The table shows the forecasting performance across different state encoding. The results exhibit best performance are highlighted in **bold** and the second best performance are underlined. Additionally, for the zero padding compared to positional encoding, the beaten performance is **highlighted**.

We conduct an ablation study based on the state encoding method. First, the construction of episodes is divided into two approaches: a non-sequential method based on random sampling and a sequential method following the conventional time axis order. Next, the composition of episode length is categorized into a dynamic approach with random sampling and a static approach using a fixed window. Note that static length is set to 1500 for Exchange and ETTh1, and 200 for Illness. As shown in Table 5, it is evident that the proposed method, non-sequential episodes with dynamic length, exhibits the most superior and robust performance, overall. Following this, the traditional approach, sequential episodes with static length, shows second robust performance. This confirms that DTF-net achieves excellent performance through the bidirectional learning it aims for, while the sequential episode with static length method occasionally exhibits overfitting results.

Next, positional encoding achieves superior performance compared to zero padding. However, in the Illness dataset, zero padding performs better. This is because Illness has a relatively short past horizon, which is 104, making simple zero padding more effective compared to Exchange and ETTh. However, as the past horizon for forecasting increases, reaching 336, positional encoding proves to be more effective.

## C.2 Ablation Study on Reward with Seed Test

| Seed | | 2023 | | 52 | | 454 | | 470 | | 515 | | 695 | | 1561 | | 1765 | | 1953 | | 2021 | | 2022 | |
|---|---|---|---|---|---|---|---|---|---|---|---|---|---|---|---|---|---|---|---|---|---|---|---|
| Metric | | MSE | MAE | MSE | MAE | MSE | MAE | MSE | MAE | MSE | MAE | MSE | MAE | MSE | MAE | MSE | MAE | MSE | MAE | MSE | MAE | MSE | MAE |
| Exchange | 24 | 0.0250 | 0.1198 | 0.0265 | 0.1236 | 0.0256 | 0.1211 | 0.0256 | 0.1220 | 0.0258 | 0.1209 | 0.0267 | 0.1237 | 0.0261 | 0.1226 | 0.0253 | 0.1203 | 0.0267 | 0.1239 | 0.0263 | 0.1227 | 0.0265 | 0.1231 |
| | 48 | 0.0487 | 0.1658 | 0.0499 | 0.1682 | 0.0498 | 0.1677 | 0.0503 | 0.1691 | 0.0492 | 0.1658 | 0.0492 | 0.1674 | 0.0500 | 0.1681 | **0.0480** | **0.1655** | 0.0493 | 0.1680 | **0.0480** | **0.1650** | 0.0510 | 0.1703 |
| | 96 | 0.0983 | 0.2349 | 0.1007 | 0.2363 | 0.0989 | 0.2349 | **0.0970** | **0.2322** | 0.0985 | 0.2322 | 0.0982 | 0.2356 | 0.0992 | 0.23540 | 0.0994 | 0.2360 | 0.0982 | 0.2350 | 0.1007 | 0.2335 | 0.1004 | 0.2368 |
| | 192 | 0.1983 | 0.3583 | 0.2096 | 0.3673 | 0.2004 | 0.3562 | 0.2023 | 0.3565 | 0.2115 | 0.3751 | 0.2009 | 0.3567 | **0.1978** | 0.3595 | 0.2063 | 0.3667 | **0.1904** | **0.3513** | 0.2056 | 0.3649 | **0.1929** | **0.3512** |
| | 336 | 0.3160 | 0.4561 | 0.4085 | 0.4939 | 0.3311 | 0.4712 | 0.3288 | 0.4678 | 0.3409 | 0.4716 | 0.3180 | 0.4467 | 0.3199 | 0.4587 | 0.3559 | 0.4874 | 0.3333 | 0.4626 | **0.3124** | **0.4578** | 0.3606 | 0.4768 |
| | 720 | 0.7933 | 0.6874 | **0.7583** | **0.6847** | 0.8022 | 0.7083 | 0.8888 | 0.7331 | 1.0462 | 0.7938 | 0.9455 | 0.7649 | 0.9338 | 0.7445 | 1.0001 | 0.7788 | 0.8769 | 0.8769 | 0.9480 | 0.7534 | **0.7889** | 0.6985 |
| ETTh1 | 24 | 0.0253 | 0.1205 | 0.0258 | 0.1219 | 0.0259 | 0.1218 | 0.0266 | 0.1234 | 0.0259 | 0.1222 | **0.0250** | **0.1197** | 0.0261 | 0.1224 | 0.0264 | 0.1235 | 0.0263 | 0.1233 | 0.0258 | 0.1214 | 0.0260 | 0.1221 |
| | 48 | 0.0375 | 0.1479 | 0.0381 | 0.1485 | 0.0383 | 0.1489 | 0.0388 | 0.1501 | 0.0404 | 0.1546 | 0.0388 | 0.1497 | 0.0391 | 0.1507 | 0.0389 | 0.1505 | **0.0370** | **0.1467** | 0.0391 | 0.1508 | 0.0386 | 0.1490 |
| | 96 | 0.0519 | 0.1740 | 0.0528 | 0.1763 | 0.0561 | 0.1824 | 0.0532 | 0.1774 | 0.0536 | 0.1774 | 0.0540 | 0.1780 | 0.0544 | 0.1789 | 0.0525 | 0.1755 | 0.0531 | 0.1768 | 0.0539 | 0.1777 | 0.0536 | 0.1783 |
| | 192 | 0.0676 | 0.2013 | 0.0697 | 0.2041 | 0.0683 | 0.2036 | 0.0704 | 0.2057 | 0.0698 | 0.2048 | 0.0703 | 0.2054 | 0.0681 | 0.2024 | 0.0695 | 0.2034 | 0.0695 | 0.2034 | 0.0694 | 0.2044 | 0.0686 | 0.2032 |
| | 336 | 0.0803 | 0.2247 | 0.0856 | 0.2325 | **0.0781** | **0.2231** | 0.0818 | 0.2272 | 0.0834 | 0.2285 | 0.0833 | 0.2291 | **0.0801** | **0.2247** | 0.0792 | 0.2232 | 0.0792 | 0.2255 | 0.0819 | 0.2278 | 0.0807 | 0.2253 |
| | 720 | 0.0776 | 0.2224 | 0.0784 | 0.2239 | 0.0823 | 0.2288 | 0.0820 | 0.2285 | 0.0832 | 0.2304 | 0.0820 | 0.2278 | 0.0813 | 0.2278 | 0.0814 | 0.2282 | 0.0792 | 0.2255 | 0.0802 | 0.2252 | 0.0805 | 0.2262 |
| Illness | 24 | 0.5881 | 0.5358 | 0.6275 | 0.5617 | 0.6100 | 0.5498 | 0.6684 | 0.5718 | 0.6285 | 0.5689 | **0.5647** | **0.5117** | **0.5764** | **0.5310** | 0.6112 | 0.5424 | **0.5476** | **0.5303** | 0.7151 | 0.6080 | 0.6570 | 0.5621 |
| | 48 | 0.6858 | 0.6359 | 0.7178 | 0.6531 | **0.6534** | **0.6170** | 0.7407 | 0.6822 | 0.7241 | 0.6626 | **0.5881** | **0.5655** | 0.6829 | 0.6604 | 0.6343 | 0.6021 | **0.5991** | **0.5642** | 0.7104 | 0.6532 | 0.7159 | 0.6435 |
| | 60 | 0.6640 | 0.6423 | 0.6682 | 0.6448 | 0.7822 | 0.7338 | 0.7314 | 0.6879 | **0.6492** | **0.6290** | 0.7194 | 0.6805 | 0.7020 | 0.6607 | **0.6526** | **0.6399** | 0.7493 | 0.6955 | 0.6769 | 0.6408 | **0.6479** | **0.6196** |

| var / length | 24 | 48 | 60 | 96 | 192 | 336 | 720 |
|---|---|---|---|---|---|---|---|
| Exchange | $3e-7$ | $9e-7$ | - | $1e-6$ | $5e-5$ | $8e-4$ | $9e-3$ |
| ETTh1 | $1e-7$ | $7e-7$ | - | $1e-6$ | $6e-7$ | $5e-6$ | $2e-6$ |
| Illness | $3e-3$ | $3e-3$ | $2e-3$ | - | - | - | - |

Table 6: **Seed test result with interval reward.** The table shows the forecasting performance across various seeds. The performance is evaluated using DTF-net with rewards sampled at equal intervals. The results from the seed that exhibited better performance, with 2023 as the reference seed, are highlighted in **bold**. The second table shows the performance variance of the seed tests based on MSE.

| Seed | | 2023 | | 52 | | 454 | | 470 | | 515 | | 695 | | 1561 | | 1765 | | 1953 | | 2021 | | 2022 | |
|---|---|---|---|---|---|---|---|---|---|---|---|---|---|---|---|---|---|---|---|---|---|---|---|
| Metric | | MSE | MAE | MSE | MAE | MSE | MAE | MSE | MAE | MSE | MAE | MSE | MAE | MSE | MAE | MSE | MAE | MSE | MAE | MSE | MAE | MSE | MAE |
| Exchange | 24 | 0.0250 | 0.1198 | 0.0265 | 0.1237 | 0.0251 | 0.1198 | 0.0259 | 0.1228 | 0.0250 | 0.1186 | 0.0254 | 0.1207 | 0.0254 | 0.1205 | 0.0257 | 0.1210 | 0.0269 | 0.1244 | 0.0263 | 0.1230 | 0.0261 | 0.1218 |
| | 48 | 0.0487 | 0.1658 | 0.0491 | 0.1666 | 0.0498 | 0.1678 | 0.0502 | 0.1682 | 0.0492 | 0.1658 | 0.0491 | 0.1665 | 0.0497 | 0.1670 | 0.0498 | 0.1652 | 0.0485 | 0.1665 | 0.0498 | 0.1683 | | |
| | 96 | 0.0983 | 0.2349 | 0.1003 | 0.2356 | 0.1009 | 0.2348 | 0.0970 | 0.2322 | 0.0968 | 0.2308 | 0.0952 | 0.2323 | 0.0985 | 0.2345 | 0.0991 | 0.2355 | 0.0996 | 0.2366 | 0.1006 | 0.2343 | 0.1004 | 0.2366 |
| | 192 | 0.1983 | 0.3583 | 0.2091 | 0.3652 | 0.2019 | 0.3575 | 0.1994 | 0.3538 | 0.2130 | 0.3760 | 0.2028 | 0.3579 | 0.1990 | 0.3593 | 0.2093 | 0.3693 | 0.1907 | 0.3515 | 0.2055 | 0.3648 | 0.1924 | 0.3508 |
| | 336 | 0.3160 | 0.4561 | 0.4040 | 0.4926 | 0.3329 | 0.4728 | 0.3284 | 0.4675 | 0.3409 | 0.4719 | 0.3202 | 0.4480 | 0.3193 | 0.4581 | 0.3571 | 0.4880 | 0.3334 | 0.4629 | 0.3122 | 0.4585 | 0.3638 | 0.3638 |
| | 720 | 0.7933 | 0.6874 | 0.7600 | 0.6855 | 0.7693 | 0.6920 | 0.8888 | 0.7332 | 1.0404 | 0.7913 | 0.9442 | 0.7636 | 0.9398 | 0.7483 | 0.9685 | 0.7745 | 0.8746 | 0.7249 | 0.9457 | 0.7529 | 0.7856 | 0.6955 |
| ETTh1 | 24 | 0.0253 | 0.1205 | 0.0257 | 0.1214 | 0.0258 | 0.1218 | 0.0268 | 0.1235 | 0.0257 | 0.1218 | **0.0250** | **0.1196** | 0.0261 | 0.1225 | 0.0261 | 0.1222 | 0.0254 | 0.1203 | 0.0258 | 0.1215 | 0.0261 | 0.1222 |
| | 48 | 0.0375 | 0.1479 | 0.0381 | 0.1485 | 0.0386 | 0.1493 | 0.0396 | 0.1513 | 0.0378 | 0.1488 | 0.0389 | 0.1499 | 0.0391 | 0.1506 | 0.0387 | 0.1502 | 0.0370 | 0.1470 | 0.0388 | 0.1502 | 0.0385 | 0.1490 |
| | 96 | 0.0519 | 0.1740 | 0.0529 | 0.1765 | 0.0557 | 0.1819 | 0.0535 | 0.1779 | 0.0537 | 0.1778 | 0.0525 | 0.1751 | 0.0543 | 0.1789 | 0.0526 | 0.1757 | 0.0525 | 0.1760 | 0.0538 | 0.1774 | 0.0536 | 0.1783 |
| | 192 | 0.0676 | 0.2013 | 0.0698 | 0.2040 | 0.0709 | 0.2067 | 0.0717 | 0.2081 | 0.0698 | 0.2049 | 0.0697 | 0.2046 | 0.0680 | 0.2023 | 0.0696 | 0.2034 | 0.0703 | 0.2063 | 0.0695 | 0.2042 | 0.0690 | 0.2040 |
| | 336 | 0.0803 | 0.2247 | 0.0845 | 0.2314 | **0.0783** | **0.2229** | 0.0818 | 0.2272 | 0.0825 | 0.2277 | 0.0844 | 0.2302 | 0.0800 | 0.2221 | 0.0800 | 0.2268 | 0.0801 | 0.2253 | 0.0804 | 0.2271 | 0.0800 | 0.2253 |
| | 720 | 0.0776 | 0.2224 | 0.0782 | 0.2234 | 0.0808 | 0.2267 | 0.0826 | 0.2292 | 0.0823 | 0.2292 | 0.0790 | 0.2237 | 0.0808 | 0.2272 | 0.0801 | 0.2253 | 0.0805 | 0.2271 | 0.0804 | 0.2254 | 0.0800 | 0.2253 |
| Illness | 24 | 0.5881 | 0.5358 | 0.6157 | 0.5495 | 0.6326 | 0.5689 | 0.6237 | 0.5672 | 0.7393 | 0.6669 | **0.6641** | **0.5293** | 0.6009 | 0.5684 | 0.6278 | 0.5575 | 0.6121 | 0.5471 | 0.6330 | 0.5590 | 0.6681 | 0.5958 |
| | 48 | 0.6858 | 0.6359 | 0.7181 | 0.6537 | 0.7076 | 0.6407 | 0.7800 | 0.7061 | 0.7205 | 0.6726 | **0.6398** | **0.5907** | 0.6958 | 0.6621 | **0.6363** | **0.6025** | 0.6254 | 0.5743 | **0.6340** | **0.6171** | 0.7155 | 0.6495 |
| | 60 | 0.6640 | 0.6423 | **0.6542** | **0.6329** | 0.7767 | 0.7326 | 0.7394 | 0.6881 | 0.6728 | 0.6501 | 0.7359 | 0.6929 | 0.7140 | 0.6658 | 0.6626 | 0.6460 | 0.7291 | 0.6829 | **0.6598** | **0.6236** | 0.6483 | 0.6203 |

| var / length | 24 | 48 | 60 | 96 | 192 | 336 | 720 |
|---|---|---|---|---|---|---|---|
| Exchange | $4e-7$ | $5e-7$ | - | $4e-7$ | $5e-7$ | $8e-4$ | $9e-3$ |
| ETTh1 | $2e-7$ | $5e-7$ | - | $1e-6$ | $1e-6$ | $4e-6$ | $2e-6$ |
| Illness | $2e-3$ | $3e-3$ | $2e-3$ | - | - | - | - |

Table 7: **Seed test result with random reward.** The table shows the forecasting performance across various seeds. The performance is evaluated using DTF-net with rewards sampled at random intervals. The results from the seed that exhibited better performance, with 2023 as the reference seed, are highlighted in **bold**. The second table shows the performance variance of the seed tests based on MSE.

We perform experiments under two conditions to evaluate the performance of DTF-net using different seeds: 1) sampling rewards at equal intervals (Table 6), and 2) randomly sampling rewards (Table 7), using 10 randomly selected seeds. The outcomes presented in Tables 6 and 7 reveal a limitation associated with high variance, stemming from the sensitivity of RL's hyperparameters and the choice of seeds. However, it's worth noting that there is also an advantage in that by identifying optimal hyperparameters and seeds, better performance can be achieved.

# D  LIMITATION OF DTF-NET

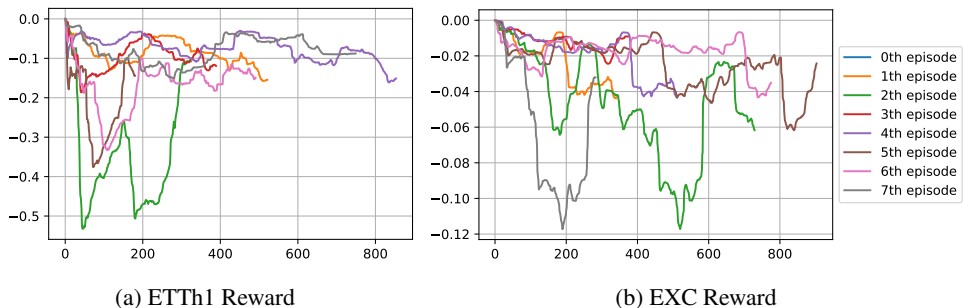

(a) ETTh1 Reward          (b) EXC Reward

Figure 7: The figure shows rewards obtained from dynamically segmented sub-sequences across different episodes from ETTh1 and exchange rate (EXC). Since DTF-net uses the forecasting cost function as a reward, the reward tends to be unstable.

## D.1  DTF-NET ON STATIONARY TSF: WEATHER AND TRAFFIC DATASET

| Methods | | DTF-Linear (ours) | | $\ell_1(\lambda = 0.1)$-Linear | | NLinear | | DLinear | | FEDformer-f | | FEDformer-w | | Autoformer | |
|---|---|---|---|---|---|---|---|---|---|---|---|---|---|---|---|
| Metric | | MSE | MAE | MSE | MAE | MSE | MAE | MSE | MAE | MSE | MAE | MSE | MAE | MSE | MAE |
| Weather | 24 | 0.0004 | 0.0143 | 0.0004 | 0.0133 | 0.0004 | 0.0128 | 0.0019 | 0.0323 | 0.0272 | 0.1263 | 0.0217 | 0.0217 | 0.0133 | 0.0957 |
| | 48 | 0.0008 | 0.0208 | 0.0008 | 0.0188 | 0.0007 | 0.0190 | 0.0043 | 0.0513 | 0.0053 | 0.0586 | 0.0055 | 0.0599 | 0.0115 | 0.0850 |
| | 96 | 0.0010 | 0.0236 | 0.0010 | 0.0227 | 0.0010 | 0.0233 | 0.0047 | 0.0543 | 0.0096 | 0.0770 | 0.0055 | 0.0593 | 0.0094 | 0.0769 |
| | 192 | 0.0012 | 0.0253 | 0.0012 | 0.0257 | 0.0012 | 0.0261 | 0.0054 | 0.0591 | 0.0048 | 0.0558 | 0.0048 | 0.0559 | 0.0055 | 0.0570 |
| | 336 | 0.0014 | 0.0277 | 0.0014 | 0.0279 | 0.0014 | 0.0278 | 0.0064 | 0.0664 | 0.0049 | 0.0554 | 0.0049 | 0.0552 | 0.0082 | 0.0683 |
| | 720 | 0.0019 | 0.0318 | 0.0019 | 0.0323 | 0.0019 | 0.0329 | 0.0066 | 0.0679 | 0.0036 | 0.0479 | 0.0036 | 0.0478 | 0.0055 | 0.0561 |
| Traffic | 24 | 0.1155 | 0.1963 | 0.1248 | 0.2174 | 0.1159 | 0.1962 | 0.1166 | 0.1986 | 0.1526 | 0.2535 | 0.1506 | 0.2432 | 0.2279 | 0.3461 |
| | 48 | 0.1251 | 0.2110 | 0.1327 | 0.2240 | 0.1214 | 0.2002 | 0.1228 | 0.3505 | 0.1729 | 0.2772 | 0.1803 | 0.2759 | 0.2523 | 0.3666 |
| | 96 | 0.1391 | 0.2283 | 0.1402 | 0.2322 | 0.1282 | 0.2074 | 0.1300 | 0.2114 | 0.1890 | 0.2884 | 0.1933 | 0.2872 | 0.2550 | 0.3665 |
| | 192 | 0.1389 | 0.2263 | 0.1429 | 0.2354 | 0.1328 | 0.2132 | 0.1331 | 0.2151 | 0.1901 | 0.2936 | 0.1955 | 0.2978 | 0.2531 | 0.3594 |
| | 336 | 0.1619 | 0.2629 | 0.1419 | 0.2377 | 0.1301 | 0.2163 | 0.1331 | 0.2213 | 0.1980 | 0.3073 | 0.2000 | 0.3092 | 0.2965 | 0.3926 |
| | 720 | 0.1548 | 0.2518 | 0.1550 | 0.2516 | 0.1423 | 0.2283 | 0.1455 | 0.2349 | 0.2601 | 0.3469 | 0.2634 | 0.3474 | 0.3935 | 0.4562 |

Table 8: We conduct TSF experiments using three datasets: Weather and Traffic. We evaluate performance using MSE and MAE, where lower values indicate better performance. In the following results, the best-performing models using DTF-net are highlighted in **bold**, and models using $\ell_1$ trend filtering are highlighted in *italic*. To compare the results, the best-performing models using the original data are underlined.

**Limitation**  DTF-net is primarily designed to address the trend filtering problem, with a specific focus on capturing abrupt changes driven by extreme values. Consequently, DTF-net may not be the most suitable choice for handling stationary datasets. To support this claim, we observed that both DTF-net and traditional trend filtering exhibit suboptimal performance in the realm of TSF when applied to stationary datasets. This suggests that, in the context of stationary datasets, whether the emphasis is on smoothing or capturing abrupt changes, trend filtering may impede rather than enhance TSF performance. However, as indicated by the results in Table 8, we substantiate that DTF-net excels in terms of performance when dealing with non-stationary and complex datasets.

## D.2  COMPUTATIONAL COST

Trend filtering is currently being researched in various approaches, ranging from traditional algorithms that optimize a single objective function to methods that utilize deep learning (Xu et al., 2020; Khodadadi & McDonald, 2019; Wang et al., 2021a; Liu et al., 2020; Wang et al.). While directly comparing the computational cost of deep learning and traditional methods is hard, when comparing optimization and time complexity, traditional algorithms optimize a single sequence at once, resulting in $\mathcal{O}(1)$ complexity. In contrast, deep learning methods train in batches, leading to $\mathcal{O}(n)$ complexity, where $n$ is the data size. In other words, the computational cost for training and inference in deep learning increases with the data size. However, deep learning offers the advantage of higher model capacity compared to conventional methods and the ability to learn in a data-dependent manner.

The computational cost of RL can vary widely depending on several factors such as the complexity of the environment, the algorithm being used, the size of the action and state spaces, and the number of training iterations. RL algorithms involve interactions between the agent and the environment, where the agent learns to take actions to maximize cumulative rewards over time. This learning process often requires multiple iterations of trial and error.

Some RL algorithms, like Q-learning or DQN, can be computationally expensive due to their reliance on maintaining value functions or action-value tables. On the other hand, more modern algorithms like A2C, PPO, or SAC are designed to distribute the learning process across multiple parallel agents, which can significantly reduce the time required for convergence.

Additionally, the choice of neural network architecture, hyperparameters, and the size of the training dataset can also impact the computational costs. Training deep neural networks, which is common in RL, can be resource-intensive, especially if large datasets are used.

Therefore, DTF-net may have limitations on computational expenses, overfitting issues, and tuning hyper-parameters, since it's based on RL. However, DTF-net addresses these issues as follows,

- **Data-expensive:** While deep networks typically require a substantial amount of data, DTF-net demonstrates robust performance even with small synthetic datasets or Illness data, utilizing a simple MLP policy.

- **Computational-expensive:** As discussed in the Appendix, we acknowledge the high computational cost of RL. However, we structured episodes through sampling to optimize DTF-net with a minimal number of steps.

- **Overfitting:** We address the overfitting issue through reward sampling. Given that the reward we employ is a penalty with a negative value, RL learns to minimize this penalty. However, continuous application of the penalty may lead to overfitting. To mitigate this, we intermittently apply the penalty during the training of DTF-net, aiming to prevent overfitting.

- **Hyper-parameter tuning:** To minimize the number of hyperparameters and ensure the robust performance of DTF-net, we have introduced a sampling methodology.

# E    EXTENDED RELATED WORK ON RL FOR DISCRETE ACTION SPACE

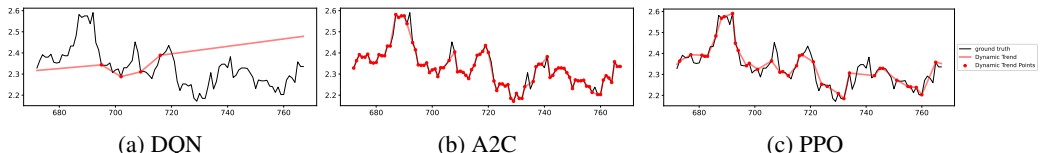

| (a) DQN | (b) A2C | (c) PPO |

Figure 8: The figure presents diverse trend-filtering outcomes achieved through various RL algorithms. Specifically, DQN exhibits an underfitting trend, whereas A2C displays an overfitting trend.

The agent performs the action $A_t$ at the state $S_t$. The environment returns $S_{t+1}$ corresponding to the following state and the reward $R_{t+1}$ to the agent. In MDP, the agent learns to predict action to maximize the cumulative reward for each state. The probability distribution that the action is selected for each state is called policy $\pi(a|s) = \Pr(A_t = a|S_t = s)$. State-value function, denoted as $v_\pi(s)$ is the expected value of return following policy $\pi$ from state $s$ as follows,

$$
\begin{aligned}
v_\pi(s) &= \mathrm{E}_\pi[G_t|S_t = s] = \mathrm{E}_\pi[R_{t+1} + \gamma v_\pi(S_{t+1})|S_t = s] \\
&= \sum_{a \in A} \pi(a|s)(r(s,a) + \gamma \sum_{s\prime \in S} p(s\prime|s,a)v_\pi(s\prime)).
\end{aligned}
\tag{4}
$$

Action-value function, denoted as $q_\pi(s, a)$ is the expected value of return from action $a$ and state $s$ as follows,

$$
\begin{aligned}
q_\pi(s,a) &= \mathrm{E}_\pi[G_t|S_t = s, A_t = a] \\
&= \mathrm{E}_\pi[R_{t+1} + \gamma q_\pi(S_{t+1}, A_{t+1})|S_t = s] \\
&= r(s,a) + \gamma \sum_{s' \in S} p(s'|s,a) \sum_{a' \in A} \pi(a'|s')q_\pi(a'|s').
\end{aligned}
\tag{5}
$$

Then, the Bellman equations express $v_\pi$ and $q_\pi$ and the optimality equations $v^*$ and $q^*$. The model that needs the state transition probability and the corresponding reward in each state, such as value-iteration, is called Model-based. The model that does not need the above information, such as Q-learning, is called Model-free (Jaderberg et al., 2016; Sutton & Barto, 2018; Mnih et al., 2013; Silver et al., 2014).

**DQN**  DQN is the first approach to solve the problems of traditional RL, such as sample correlation and change in the data distribution. The target value is defined by the following two methods. First, the Experience Replay Buffer decreases updated variance and sample correlation by random extraction. Since the Q-function considers various actions simultaneously, the policy is averaged to solve a bias in the data distribution. Second, the target network has a dual structure in the main Q-Network (Mnih et al., 2013; 2015).

**A2C**  If the episode is prolonged, the variance increases. To overcome this, Actor-Critic does not use the Replay Buffer but learns action directly. The actor-critic defines the policy network using the estimated value function (Sutton & Barto, 2018).

**PPO**  As for model-free-based learning, PPO is a variation of A2C, which reuses learned data with sampling. PPO has the advantage of using both continuous and discrete actions. PPO approximates first-order derivation using clipping to solve the complexity of the surrogate object function in TRPO. Here, TRPO is a trust region that makes stability in training. Finally, PPO optimizes the objective function by reflecting the Action Entropy for State-Value and Exploration (Schulman et al., 2015; 2017).

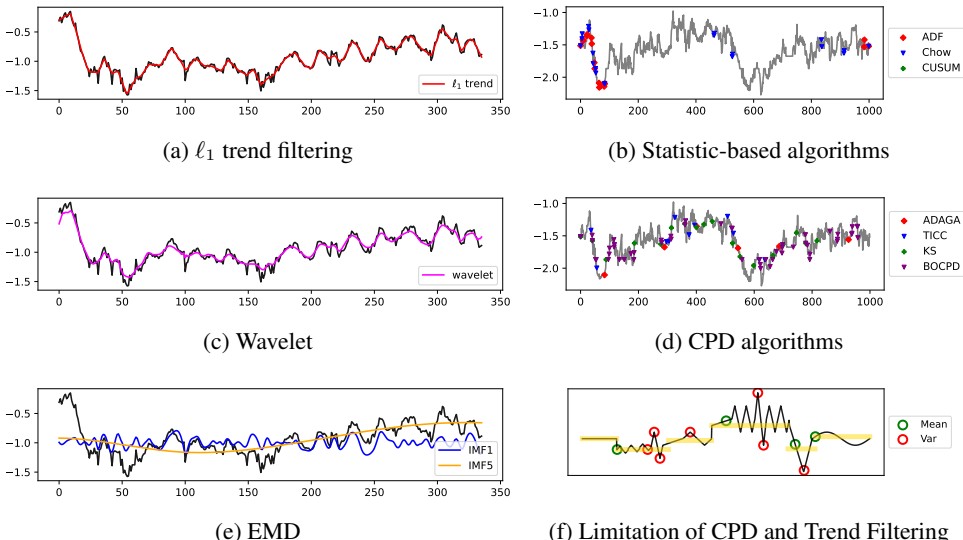

Figure 9: 1) The first-column graph shows the trend extracted from the $\ell_1$, Wavelet, and EMD methods. The constant smoothness extracts a rigid trend that filters out noise for entire sequences. 2) The second-column graph shows that statistical-based and CPD algorithms detect abrupt changes with their criteria in a real-world dataset: ETTh1. The detected points are irregular and difficult to validate. 3) Figure (f) shows the challenges with current algorithms when identifying abrupt changes, as highlighted by the red circle, through only shifts in variance.

# F   DISCUSSION

## F.1   LIMITATIONS OF TRADITIONAL TREND FILTERING ALGORITHMS

**H-P and $\ell_1$ Trend Filtering**  Hodrick-Prescott filtering (H-P) (Hodrick & Prescott, 1997) extracts trends by minimizing the residual between original data $y_t$ and trend $x_t$. Given a scalar time series $y_t \in \mathbb{R}$, where $t = \{1, ..., n\}$, assume slow-varying $x_t \in \mathbb{R}$ and abrupt-varying $(y_t - x_t)$. To estimate $x_t$, H-P trend filtering minimizes the weighted sum objective function as follows,

$$\frac{1}{2} \sum_{t=1}^{n} (y_t - x_t)^2 + \lambda \sum_{t=2}^{n-1} (x_{t-1} - 2x_t + x_{t+1})^2, \tag{6}$$

where $\lambda \in \mathbb{R}^+$ is a regularization parameter, which adjusts the degree of smoothness.

$\ell_1$ trend filtering (Kim et al., 2009) is a variation on H-P trend filtering. $\ell_1$ trend filtering estimates the smoothness in a piece-wise linear sense. To better reflect abrupt changes, $\ell_1$ trend filtering replaces the sum of squares term with a sum of the absolute values. However, the second-order difference operators cause two crucial limitations. One is delayed detection of abrupt changes, even using future time-step $x_{t+1}$. The other is that optimal $\lambda$ should be found to obtain the trend with the appropriate degree of smoothness (Moghtaderi et al., 2011). On the other hand, TV-denoising trend filtering (Chan et al., 2001), which utilizes first-order differences, focuses excessively on abrupt changes, resulting in delays in capturing slow-varying trends (Wen et al., 2019).

**Wavelet Trend Filtering**  Wavelet extract trends depend on the time and frequency domain to handle non-stationary signals. Utilizing both time and frequency extracts a more robust trend containing abrupt changes. However, choosing an orthogonal basis is a difficult problem and can lead to high levels of overfitting (Wen et al., 2019; Rhif et al., 2019; Craigmile & Percival, 2002).

**EMD**  The Empirical Mode Decomposition (EMD) (Wu et al., 2007) decomposes a time series into a finite set of oscillatory modes known as the Intrinsic Mode Functions (IMFs) (Huang et al., 1998; Moghtaderi et al., 2011; Gaci, 2016). However, EMD needs to choose a free parameter for IMF that

adjusts the smoothness of the trend. In addition, the mode mixing effect can occur and the wrong mode can limit EMD due to bias in the local mean value.

A common limitation of existing trend filtering methods is their noise filter with a fixed window, which results in the inability to accurately capture abrupt changes. Constant smoothness ignores important events in long-tail time series distributions. As shown in the first column of Figure 9, the abrupt change at the 100th point is not reflected in any of the trend filtering methods.

### F.2 LIMITATIONS OF STRUCTURAL BREAK POINTS AND CHANGE POINT DETECTION ALGORITHMS TO DETECT ABRUPT CHANGES

**Statistical Test** Structural Break Points (SBP) can be applied to a wide range of time series domains. There are various statistic-based algorithms that can validate found SBPs, such as Chow (Chow, 1960), Augmented Dickey-Fuller (Dickey & Fuller, 1979), and CUSUM (Ploberger & Krämer, 1992). However, in real-world applications, detection should precede even without prior knowledge.

Figure 9 (b) shows the results of detected SBP by different statistical algorithms based on the maximum F-value. In various characteristics of SBP, each algorithm only detects fitted SBP according to their own rules. The detected breakpoints are irregular and difficult to adapt to the trend.

**Change Point Detection** Change Point Detection (CPD) is a method for identifying abrupt changes in time series data when the probability distribution changes. While CPD algorithms aren't inherently tailored for trend filtering, there is a certain degree of overlap in their objectives with trend filtering in the perspective of capturing abrupt changes. One popular approach to CPD is Bayesian Online Change Point Detection (BOCPD) (Adams & MacKay, 2007), which uses Bayesian inference. However, BOCPD has limitations in practical applications due to its assumption of independence and the presence of temporal correlations between samples (Saatçi et al., 2010; Caldarelli et al., 2022). Additionally, the constant hazard function $H(\tau)$ makes BOCPD sensitive to hyperparameters (Han et al., 2019).

The Kolmogorov-Smirnov (KS) test (Gong & Huang, 2012) segments time series into left and right sections. Since the KS algorithm is only applicable in offline detection, the dynamic length of segmentation is extracted by considering all segment cases of the entire time series. However, offline detection is highly sensitive to the input length, and an additional threshold must be defined heuristically to detect change points. Additionally, RED-SDS (Ansari et al., 2021) needs to employ a labeling process to determine the correct change points. Segmentation algorithms can miss important change points within the segment due to the clustering effect.

The clustering-based TICC algorithm (Hallac et al., 2017) generates a sparse Gaussian inverse covariance matrix by segments, using Lasso regularization. However, since the number of clustering functions is a hyper-parameter that requires prior knowledge, only a fixed number of change points can be detected. Similar to the KS test, finding change points within the same cluster is difficult.

The Gaussian Process (GP) and Bayesian-based ADAGA algorithm (Caldarelli et al., 2022) detects mean shifts as change points, but its results depend on the kernel type used (IP and QFF). While GP approximates smoothness using mean-square differentiable functions (Banerjee & Gelfand, 2003), it has limitations in capturing non-smoothness (Luo et al., 2021). Consequently, Bayesian models are well-suited for CPD, but their applicability to trend filtering, which aims to include extreme values, remains uncertain.

The second column of Figure 9 reveals irregularities in detecting change points on the ETTh1 dataset (Zhou et al., 2021) by algorithms such as Chow, ADF, CUSUM, KS test, TICC, and BOCPD, which rely on different criteria. Furthermore, as shown in Figure 9 (f), probabilistic models (Adams & MacKay, 2007; Caldarelli et al., 2022; Ansari et al., 2021) have a significant limitation in detecting mid-points (green) instead of the vertices of abrupt changes (red). Further discussion on this topic can be found in the Appendix.

### F.3 LIMITATION OF ANOMALY DETECTION ALGORITHMS TO DETECT ABRUPT CHANGES

Anomaly detection involves identifying abnormal points or sub-sequences in time series data, aligning with the concept of extreme values. As capturing extreme values is a common objective with DTF-net, we compare recent anomaly detection algorithms regarding their ability to capture abrupt

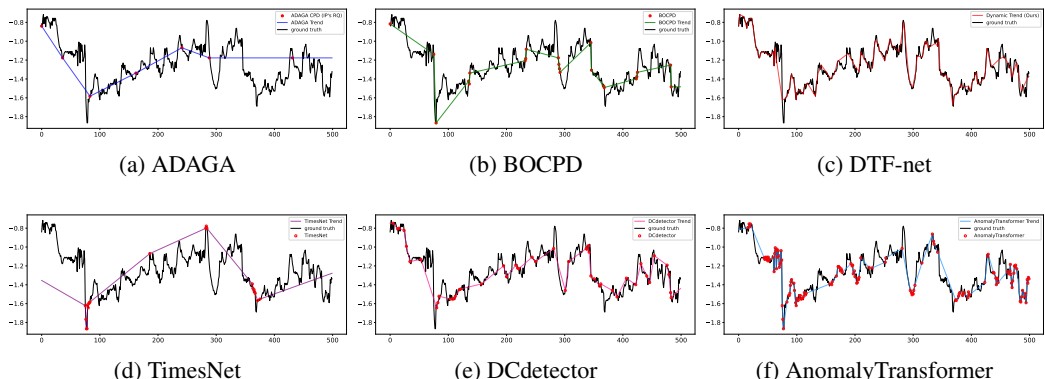

Figure 10: **A comparative experiment between Change Point Detection (CPD), Anomaly Detection, and DTF-net for trend filtering on ETTh1.** While the CPD algorithm effectively captures change points, it faces challenges in extracting trends only based on the detected points (red dots). However, DTF-net (red line) dynamically performs trend filtering by including abrupt changes, such as the 300th point.

changes. This includes assessments of TimesNet (Wu et al., 2023), DCdetector (Yang et al., 2023), and Anomaly Transformer (Xu et al., 2022).

First, TimesNet employs a Temporal Variation Modeling methodology to transform the original 1D time series into a 2D space, unifying intra-period and inter-period variations. However, it tends to underfit, capturing only extreme anomalies and making trend extraction challenging. Next, DC detector adopts a novel dual attention asymmetric design, creating a permutated environment and using pure contrastive loss for the learning process. While effective in capturing anomalies, DC detector is weak in handling complex sub-sequences, such as those around axis 80. Finally, Anomaly Transformer introduces a new Anomaly-Attention mechanism to compute association discrepancies, applying a minimax strategy to enhance normal-abnormal distinguishability. Although Anomaly Transformer closely performs trend filtering as anomaly detection, it misses important peaks like those at 150, 250, and 450. In conclusion, anomaly detection results in varying smoothness of extracted trends depending on the threshold, and even with an appropriate threshold, points are often missed in peak regions.

### F.4 TREND FILTERING AND ABRUPT CHANGE POINT

Existing trend filtering research has primarily focused on capturing abrupt changes in trends. More recently, there has been a growing interest in utilizing deep learning models for trend filtering (Wang et al., 2021a; Liu et al., 2020; Wang et al., 2021b; Xu et al., 2020). However, a common drawback observed in these studies is their still failure to effectively capture abrupt changes in the data. Abrupt changes correspond to points in the time series where the trend's slope experiences significant variations, often indicating important events. To overcome this limitation, we propose a novel approach that specifically targets abrupt changes in trend filtering. DTF-net is an innovative methodology that leverages RL techniques grounded in EVT to detect trend points, including abrupt changes, enabling dynamic trend filtering.

Since real-world time series data is non-stationary and chaotic, it is challenging to extract trends using a fixed criterion or a single distribution. Consequently, recent methods have been proposed to address the Out-Of-Distribution (OOD) challenge (Lu et al., 2022; Belkhouja et al., 2022). However, finding the optimal number of sub-distributions is difficult, and dividing the entire sequence into numerous distributions is not a fundamental solution. This approach can lead to overfitting due to the limited number of distributions in the dataset and make it challenging to learn invariant features from sub-series (Lu et al., 2022).

In contrast, our objective is not only to identify change points between sub-sequences with different distributions but also to incorporate important extreme values that could be considered outliers

within the same distribution into the trend. To accomplish this, we needed to take a different approach from existing OOD methods or CPD techniques.

As illustrated in Figure 10 (a) and (b), Bayesian-based change point detection effectively captures points (red or blue) between sub-sequences with different distributions. However, constructing the trend using only these points remains challenging due to over-smoothing issues. Therefore, as depicted in (c), we employed RL to identify valid points and integrate them into the trend, thereby accurately representing the non-stationarity of the original data.

## G    FUTURE WORK

**Multivariate Trend Filtering and Time Series Forecasting**  DTF-net is applied to multivariate time series data features with the target of univariate trend filtering. This means that RL examines multidimensional aspects to perform trend filtering on the target. However, individual trend filtering for each dimension in the multidimensional case has not yet been addressed in this study. Nevertheless, this is achievable through the multi-discrete action prediction of the PPO algorithm.

