# OpenReview forum: "Towards Dynamic Trend Filtering through Trend Points Detection with Reinforcement Learning"
_ICLR.cc/2024/Conference — Submitted to ICLR 2024_

### Official Review · Reviewer_ALFQ · 2023-10-28

**Soundness:** 2 fair
**Presentation:** 2 fair
**Contribution:** 2 fair
**Rating:** 5
**Confidence:** 3

**Summary:**

This paper proposes an RL-based method to identify dynamic trend points in time series data. For a long time, the important dynamic trend points are mixed with noisy points and are hard to detect. This paper wants to use RL to identify them through a designed reward function. The experiments on synthetic data show some promising results.

**Strengths:**

The problem of this paper is interesting and important. DTPs are valuable to be identified from noise in a time series. It is a challenging problem. The authors motivate this problem well. The idea of using RL to resolve this problem is promising, even though much details are missing.

**Weaknesses:**

The main challenge for this work is to distinguish the noise points with trend points. However, there is no discussions on why RL can resolve this challenge. The authors only claim that ‘DNNs employed by RL agent is capable of capturing intricate patterns within time series data and identifying DTPs based on their non-linearity inherent.’ How do DNNs in RL can identify DTPs? Why do you think DNNs in RL can do that? What is the motivation behind that? There are no reasonable discussions on that. However, that is the main contribution of this work.

The experimental evaluations are weak. Why do the authors highlight DTF-net in Table 1 rather than the best performance? The authors claim that “Note that wavelets have an overfitting issue, resulting in insufficient noise filtering.” Have you evaluated the noise filtering? How can you show that? There are a lot of real-world time-series data. Why did the authors not evaluate the methods on them? Only one example is given in Figure 4 without the performance of other methods.

**Questions:**

Please see the Weaknesses.

---

> ### Author Response · Authors · 2023-11-19
> **Response to Reviewer ALFQ (1/)**
>
> We express our gratitude to ****Reviewer ALFQ**** for insightful feedback. In response to the reviewers’ comments, we prepared a revised version of the manuscript, and we address specific questions below.
>
> 1. How do DNNs in RL can identify DTPs? Why do you think DNNs in RL can do that?
>     1. As stated in Section 1, **when using DNN, it is unconstrained by fixed window sizes or frequencies within the time series data domain**. Additionally, deep networks learn in a **data-dependent manner**, enabling them to **capture the characteristics of sub-sequences in time series data more effectively**. This capability enables them to **perform different smoothness on abrupt changes that are distributed differently for each sub-sequence**. For example, as shown in **Figure 3 in a revised version** of the manuscript, DTF-net extracts trends with varying smoothness for each sub-sequence, in contrast to $\ell_1$. Consequently, DTF-net identified and filtered what was considered noise in the third point, while recognizing the other five points as abrupt changes.
> 2. What is the motivation behind using RL?
>     1. As stated in **Section 2.3**, our motivation stems from the **optimal stock trading** strategy, where the goal is to **buy at the lowest point and sell at the highest point, aligning with the concept of abrupt changes.** Consequently, we draw inspiration from RL stock trading algorithms to effectively capture trend points.
>     2. We supplemented our motivation in **Sections 1 and 2.3 in a revised version as follows**.
>
>     > Motivated by the nature of abrupt changes, categorized as extreme values, we introduce a novel algorithm designed to directly identify essential points for trend extraction, departing from approximations. **This is inspired by the optimal stock trading strategy aimed at maximizing profit.** Attaining the optimal strategy involves **buying at the lowest point and selling at the highest point to achieve the maximum return, and these points align with the concept of abrupt changes** and extreme values. In this context, **Reinforcement Learning (RL) can capture abrupt changes, which are optimal trading points in a single stock trading task**, by an agent. Derived from these motivations, we reformulate the Reinforcement Learning (RL) stock trading algorithm as a solution to the trend filtering problem.
>     >

---

> ### Author Response · Authors · 2023-11-19
> **Response to Reviewer ALFQ (2/)**
>
> 3. Why do the authors highlight DTF-net in Table 1 rather than the best performance? (Wavelet overfitting)
>     1. In the case of **Wavelet**, it demonstrates **superior performance in terms of simple MSE or MAE metrics when applied to the ground truth of a linear signal added with noise**. However, we conclude that it tends to overfit to noisy data for two reasons. First, **when compared with the experimental results using only a linear signal as the ground truth,** as shown in Table 1 in the revised version, **there is a significant discrepancy with the result of the linear signal added noise as the ground truth**, implying substantial overfitting. Next, as an additional indication of potential overfitting, **we introduce another crucial property of trend filtering, namely, smoothness**. To evaluate this smoothness, **we make the assumption that the added Gaussian noise, accounting for 10%, should be filtered out to consider it an appropriate level of smoothness**. Therefore, Wavelet can be considered incapable of performing effective noise filtering.
>     2. We **added the assumption of smoothness, “filtering out at least 10% of Gaussian noise”,** in a revised version and provided updated results in **Table 1**.
> 4. There are a lot of real-world time-series data. Why did the authors not evaluate the methods on them?
>     1. Analyzing trend filtering methods poses two challenges. First, **defining a ground truth for the trend is challenging**. For example, in the case of $\ell_1$, the degree of smoothness varies with $\lambda$, resulting in different trends. Second, **labeling abrupt changes is challenging**. Obtaining real-world data with such labels is difficult, and arbitrary assignment to real-world data is subjective. **Therefore, we performed performance evaluations by setting time series forecasting** as a downstream task, indirectly allowing the assessment of trend filtering results on real-world data.
>     2. We **supplemented why trend filtering analysis on real-world data poses challenges** in a revised version.
>
> We would again like to thank ****Reviewer ALFQ****, and we hope that our changes adequately address your concerns. Please let us know if you have any further questions or comments, and we are very happy to follow up!

---

> > ### Comment · Reviewer_ALFQ · 2023-11-22
> >
> > Thanks for the authors' responses!
> >
> > I understand the difficulty of finding appropriate time-series data for the given problem and why the wavelet method is claimed to be problematic.
> >
> > However, I am still concerned about the first question. I do not think the simple DNN is just able to detect different smoothness and abrupt changes. Most of DNNs are with smoothness assumption and have the property of catastrophic forgetting.
> >
> > Hence, I can only minorly increase my score.

---

> ### Author Response · Authors · 2023-11-23
> **Response to Reviewer ALFQ**
>
> Thank you for raising the score!
>
> We are grateful to hear that some of your concerns have been addressed and we additionally respond to the first question.
>
> 1. I do not think the simple DNN is just able to detect different smoothness and abrupt changes. Most of DNNs are with smoothness assumption and have the property of catastrophic forgetting.
>     1. We acknowledge that DNNs may have issues related to smoothness and forgetting, and we also mentioned in Section 2.2 that the smoothness of DNN is approximated to a Gaussian output by the MSE loss function. **Therefore, in this paper, rather than directly extracting trend filtering from the output of DNN, which is a regression task, we train the policy of an RL agent to capture trend points through actions.** Consequently, we believe that the issues related to DNN's smoothness assumption or forgetting may not be problematic.
>     2. Additionally, we conceptualized DTF-net to identify trend points by considering individual sub-sequences in the environment. This is achieved through a forecasting task serving as the reward. Moreover, as illustrated in Figure 3, we demonstrated that DTF-net displays distinct smoothness for each sub-sequence in real-world data.
>
> If you have any further questions or comments regarding the first question, we would be happy to follow up!

---

### Official Review · Reviewer_Fg2U · 2023-10-30

**Soundness:** 2 fair
**Presentation:** 3 good
**Contribution:** 2 fair
**Rating:** 5
**Confidence:** 3

**Summary:**

In this paper, the authors introduce a trend filtering approach, called DTF-net, to capture abrupt changes in time series through reinforcement learning methods. Instead of filtering them out as what many traditional trend filtering approaches do, the approach call such abrupt changes as Dynamic Trend Points (DTP), and formulate DTP detection problem as a Markov Decision Process. Subsequently, the reinforcement learning conducts MSE loss as reward function, so as to capture the temporal dependency simultaneously. The results demonstrate its ability to capture abrupt changes.

**Strengths:**

- The authors propose to formulate DTP detection problem as a Markov Decision Process, and utilize reinforcement learning to tackle it, which is seldom explored in trend filtering problem.
- Their opinion of detecting the abrupt changes instead of filtering them out is feasible, and their experiment of including the trend as an additional input feature for Time Series Forecasting also improves model's performance across several datasets.

**Weaknesses:**

1. Section 3.1.2

In Section 3.1.2, the authors fail to provide a satisfactory explanation or analysis for the superiority of random sampling over traditional sequential sampling. They briefly mention the challenge of dynamic-length sampling and their use of state encoding to maintain a constant state length. It's worth considering whether dynamic-length sampling is necessary or if equal-length sampling would suffice. Furthermore, clarity is needed on how they determine different lengths for various sampling samples. Additionally, the claim that the proposed Positional Encoding is "specifically designed for time series data" appears overstated, as it resembles the positional encoding used in the vanilla Transformer and its variants seen in other TSF Transformers like Informer or Autoformer.

2. Section 3.2

The choice of MSE loss as the reward function raises questions. It's unclear why an MSE-based reward ensures effective learning of abrupt changes rather than favoring smoother results, which could lead to lower overall MSE. Providing a more detailed rationale for this choice would be beneficial. Additionally, while the authors introduce reward sampling as a solution to overfitting, its operation and how it addresses overfitting remain unclear.

3. Experiment

An important consideration is whether this method is suitable for multivariate time series forecasting tasks, enabling the detection of abrupt change points across multiple channels simultaneously.

**Questions:**

+ The authors could give a more detailed explanation or analysis of why random sampling is better, and if dynamic-length sampling works better than traditional sequential and equal-length sampling.
+ The authors could explain how reward sampling works more detailed, and why MSE loss as reward function can help the model effectively learn the information of abrupt changes.
+ The authors could conduct more experiments on multi-variate time series if possible or state the challenges for that.

---

> ### Author Response · Authors · 2023-11-19
> **Response to Reviewer Fg2U (1/)**
>
> #
>
> We express our gratitude to ****Reviewer Fg2U**** for insightful feedback. In response to the reviewers’ comments, we prepared a revised version of the manuscript, and we address specific questions below.
>
> 1. Section 3.1) Explanation or analysis for the superiority of random sampling over traditional sequential sampling (also aligned with Q1), and is dynamic-length sampling necessary or is equal-length sampling would suffice? (also aligned with Q1)
>     1. First, we introduce the revised version of Appendix C.1.
>
>     > We conduct an ablation study based on the state encoding method. First, the **construction of episodes is divided into two approaches: a non-sequential method based on random sampling and a sequential method following the conventional time axis order**. Next, the **composition of episode length is categorized into a dynamic approach with random sampling and a static approach using a fixed window**. Note that static length is set to 1500 for Exchange and ETTh1, and 200 for Illness. As shown in Table 5, it is evident that the proposed method, **non-sequential episodes with dynamic length, exhibits the most superior and robust performance**, overall. **Following this, the traditional approach, sequential episodes with static length, shows second robust performance**. This confirms that DTF-net achieves excellent performance through the bidirectional learning it aims for, while the sequential episode with static length method occasionally exhibits overfitting results.
>     >
>
>     2. We observed that DTF-net, with its non-sequential episode progression and dynamic length, **achieves robust performance through bidirectional learning** compared to sequential learning with static length. Additionally, it **effectively mitigates overfitting issues** through the sampling approach.
>
>     3. We supplemented the detailed ablation study and its results in the revised version of  Appendix C.1.
>
> 2. Section 3.1) Clarity is needed on how they determine different lengths for various sampling methods.
>     1. We set the starting point and length of the episode's sub-sequence through random sampling from a discrete uniform distribution, as follows,
>
>         $$s \sim \text{unif}(0, N),$$
>         $$l \sim \text{unif}(h+p, H).$$
>
>     2. The reason for using a discrete uniform distribution is to ensure that all sequences are considered uniformly.
>     3. We supplemented additional detailed explanations with formulas for random sampling **in a revised version of Section 3.**
> 3. Section 3.1) Positional Encoding is "specifically designed for time series data" appears overstated.
>     1. We acknowledge that the positional encoding statement is overstated and revised it (Section 3.1). Also, we added an **ablation study on positional encoding compared to simple zero padding** methods **in a revised version of Appendix C.1 (Table 5)**.

---

> ### Author Response · Authors · 2023-11-19
> **Response to Reviewer Fg2U (2/)**
>
> 4. Section 3.2) The reason for the choice of MSE loss at the reward function.
>     1. First, we introduce the revised version of Section 3.2.
>
>     > In the stock trading task, RL optimizes based on cumulative return as the reward. However, **in general time series, we cannot specify return as a reward.** As a result, similar to traditional trend filtering, **DTF-net aims to preserve the `closeness to the original data' property**, fundamental to trend filtering. However, employing a sum of squared loss function on current values makes it challenging to reflect appropriate smoothness. Therefore, DTF-net distinguishes itself by optimizing for future values instead of current ones. **By employing Time Series Forecasting (TSF) and training RL to minimize its cost function, DTF-net can reflect the characteristics of each sub-sequence and learn temporal dependencies.** This allows adjusting smoothness by tuning the prediction window.
>     >
>
>     2. In this context, the **MSE cost function is not solely used but in conjunction with the downstream task of forecasting.** Given that time series forecasting involves learning patterns within time series data, DTF-net is capable of predicting trend points while **taking into account the unique characteristics of sub-sequences**. Additionally, in forecasting tasks, DTF-net predicts trend points while **considering temporal dependencies around abrupt changes (Table 1)**. This is the reason why we chose forecasting MSE cost function as a reward.
>
> 5. Section 3.2) Reward sampling’s operation and how it addresses overfitting.
>     1. First, we introduce the revised version of Section 3.2.
>
>     > In here, we found that by using a penalty reward as a negative value, irregularly applying penalties through sampling can prevent overfitting **rather than penalizing at every moment** (Appendix C.2).
>     >
>
>     2. Since the reward we use is a penalty with a negative value, RL learns in the direction of reducing the penalty. However, **if the penalty is consistently applied, there is a tendency for the model to overfit to the data**. Therefore, **we intermittently apply the penalty to train DTF-net**, aiming to prevent overfitting.

---

> ### Author Response · Authors · 2023-11-19
> **Response to Reviewer Fg2U (3/)**
>
> 6. Experiments) Multivariate time series forecasting
>     1. First, we introduce the revised version of the Appendix G.
>
>     > DTF-net is applied to multivariate time series data features with the target of univariate trend filtering. This means that RL examines multidimensional aspects to perform trend filtering on the target. However, **individual trend filtering for each dimension in the multidimensional case has not yet been addressed in this study**. Nevertheless, **this is achievable through the multi-discrete action prediction of the PPO algorithm.**
>     >
>
>     2. We added multivariate time series forecasting as future work in a revised version of **Appendix G**.
>
>
> We would again like to thank ****Reviewer Fg2U****, and we hope that our changes adequately address your concerns. Please let us know if you have any further questions or comments, and we are very happy to follow up!

---

> ### Author Response · Authors · 2023-11-23
> **Response to Reviewer Fg2U (4/)**
>
> We additionally respond to the second question raised by the reviewer.
>
> The selection of the MSE loss function as a reward is motivated by the $\ell_1$ trend filtering, a widely employed method in time series data. Formally, the $\ell_1$ trend filtering minimizes the following objective function,
>
> $\sum (y_t - x_t)^2 + \lambda \sum |x_{t-1} - 2x_t + x_{t+1}|$.
>
> The first term represents the proximity to the original data, while the second term represents smoothness. Therefore, to capture the proximity property of trend filtering, we utilize the first term as a reward function.
>
> Second, regarding smoothness, $\ell_1$ employs proximity to past-current value and current-future value.
>
> $|x_{t-1} - 2x_t + x_{t+1}| = |(x_{t-1} - x_t) + (x_{t+1} - x_t)|$
>
> Contrarily, we utilize the proximity of future values based on the prediction, without the additional regulation term, thereby employing forecasting MSE as a reward. We can adjust smoothness by forecasting the window as a hyperparameter as shown in Figure 6.
>
> We acknowledge the tendency of the forecasting method to lower MSE by generating a smooth output. However, as we leverage the trend extracted through the actions of DTF-net as an additional feature, we take advantage of the characteristic illustrated in Figure 2, where DTF-net performs well in capturing abrupt changes, causing a decrease in MSE. This is further supported by the evidence in Figures 4-(b) and 5, where improved performance in forecasting MSE is observed when the trend effectively reflects abrupt changes.
>
> If you have any further questions or comments regarding the second question, we would be happy to follow up!
>
> Thank you.

---

> > ### Comment · Reviewer_Fg2U · 2023-11-23
> > **Thanks for your detailed responses**
> >
> > Dear authors,
> >
> > Thank you for your detailed responses. After reading the other reviewers' comment and the responses, I decide to keep my initial score.
> >
> > Thanks again.
> >
> > Sincerely,
> >
> > Reviewer

---

### Official Review · Reviewer_NMhu · 2023-10-31

**Soundness:** 3 good
**Presentation:** 3 good
**Contribution:** 3 good
**Rating:** 8
**Confidence:** 3

**Summary:**

This paper studies time series trend filtering problem and aims to filter noise but leave important abrupt changes unchanged. The authors proposed a deep model to detect extreme values that fall into the tail part of the data distribution. Specifically, the authors try to identify essential points termed dynamic trend points (DTPs) and reflect DTPs in the trend. The identifying problem is formulated as Markov decision process (MDP) and reinforcement learning is used to address it. MSE loss is used as reward of the agent's actions. Experiments on both synthetic and real-world datasets demonstrated that the proposed algorithm can enhance time series forecasting task performance. The authors analyzed many aspects of the proposed model including both pros and limitations.

**Strengths:**

S1. The detection problem is innovatively transformed into a Markov decision process and addressed by reinforcement learning: the deep model takes time series sub-sequence as input, conducts a series of actions, and then makes forecasting (concentrating on future but not current readings). The mean squared error (MLS) is utilized as loss/reward.

S2. A random sampling scheme is proposed to enable the DTF-net reading time series from both forward and backward directions, it can also alleviate over-fitting issue from my perspective.

S3. Effectiveness is proved on a synthetic dataset and a real-world dataset. The proposed model also enhances time series forecasting task performance.

**Weaknesses:**

W1. Training a deep net might be data- and computational-expensive and may face over-fitting problem. Tuning hyper-parameters is also challenging depending on the data and real-world applications.

W2. The heuristic random sampling algorithm can be optimized by considering the environment/context of the time series trend.

W3. The overall framework might be built without reinforcement learning, which I suggest the authors explore further.

Minors:

M1. Figure 1 should be referred to in the main text.

M2. Figure 4 is not straightforward.

M3. Legend of Figure 5 is too small, unreadable.

**Questions:**

You can respond to the weaknesses if you have new insights. Though they are not questions.

---

> ### Author Response · Authors · 2023-11-19
> **Response to Reviewer NMhu (1/)**
>
> We express our gratitude to ****Reviewer NMhu**** for positive and insightful feedback. In response to the reviewers’ comments, we prepared a revised version of the manuscript, and we address specific questions below.
>
> 1. W1) Computational expensive, overfitting problem, and tuning hyper-parameters.
>     1. We acknowledge that the limitations of DTF-net include data and computational expenses, the risk of overfitting, and the need for hyperparameter tuning. However, we have made efforts to address or have addressed these issues in the following ways.
>     2. **Data-expensive**: While deep networks typically require a substantial amount of data, DTF-net demonstrates robust performance even with small synthetic datasets or Illness data, utilizing a simple MLP policy.
>     3. **Computational-expensive**: As discussed in the Appendix, we acknowledge the high computational cost of RL. However, we structured episodes through sampling to optimize DTF-net with a minimal number of steps.
>     4. **Overfitting**: We address the overfitting issue through reward sampling. Given that the reward we employ is a penalty with a negative value, RL learns to minimize this penalty. However, continuous application of the penalty may lead to overfitting. To mitigate this, we intermittently apply the penalty during the training of DTF-net, aiming to prevent overfitting.
>     5. **Hyper-parameter tuning**: To minimize the number of hyperparameters and ensure the robust performance of DTF-net, we have introduced a sampling methodology.
>     6. We supplemented these discussions in a revised version of **Appendix D.2**.
> 2. W2) Heuristic random sampling algorithm, and W3) Overall framework might be built without RL.
>     1. Thank you for your suggestions. Specifically, we found methods that solve MDP without RL, including, Dynamic Programming, Linear Programming, Policy Gradient Methods, Monte Carlo Methods, Q-Learning (Model-Free), and Temporal Difference. We plan to add these methods in Appendix G, Future Work section. If the methods that the reviewer wants to specify are not added, we want to hear from you through discussion.
> 3. Minors
>     1. M1) Figure 1 should be referred to in the main text.
>     2. M2) Figure 4 is not straightforward.
>     3. M3) Legend of Figure 5 is too small, unreadable.
>         1. Thank you for bringing these to our attention. We addressed these minor points in the revised version.
>
> We would again like to thank ****Reviewer NMhu****, and we hope that our changes adequately address your concerns. Please let us know if you have any further questions or comments, and we are very happy to follow up!

---

### Official Review · Reviewer_Q1zH · 2023-11-01

**Soundness:** 2 fair
**Presentation:** 3 good
**Contribution:** 2 fair
**Rating:** 6
**Confidence:** 3

**Summary:**

The inherent smoothness of trend filtering filters out the tail distribution of time series data, characterized as extreme values, thereby failing to reflect abrupt changes in the trend.  In this work, they formalize the Trend Point Detection problem as a Markov Decision Process (MDP) to Dynamic Trend Points (DTPs). And they solve the Trend Point Detection problem using Reinforcement Learning (RL) algorithms operating within a discrete action space, referred to as the Dynamic Trend Filtering network (DTF-net). And they finally demonstrate that DTF-net excels at capturing abrupt changes compared to traditional trend filtering and also enhances performance in forecasting tasks.

**Strengths:**

This paper analyses the problem that the smoothing of Traditional trend filtering may also filter out abrupt changes that should be manifested in the trend, thus capturing abrupt changes through the proposed DTF-net. I think the origin of the problem is very clear and It contributes to the field.
The authors define the essential points in the trend as Dynamic Trend Points (DTPs), and the process of capturing them is referred to as Trend Point Detection. And formalize the Trend Point Detection problem as a Markov Decision Process (MDP), so as to address the issue using reinforcement learning methods.  So I think the formal transformation of the problem is innovative and the solution and ideas used are appropriate.

**Weaknesses:**

I believe that the problem analysis and theoretical exposition in this paper are sufficient, but there is still room for enrichment in the methods sections.  For example, you can formalize the reinforcement learning process with diagrams to enrich the details.

**Questions:**

I noticed that the baselines of several recent sotas compared in your experiment are all geared towards trend filtering, and I'm wondering if their methods take abrupt changes detection into account, and if there are other methods such as anomaly detection that can be used instead of, or to help, produce similar results.

---

> ### Author Response · Authors · 2023-11-19
> **Response to Reviewer Q1zH (1/)**
>
> We express our gratitude to ****Reviewer Q1zH**** for positive and insightful feedback. In response to the reviewers’ comments, we prepared a revised version of the manuscript, and we address specific questions below.
>
> 1. Formalize the reinforcement learning process with diagrams to enrich the details.
>     1. Thank you for your suggestion and **we added a diagram in a revised version of Figure 1.**
> 2. Compared trend filtering methods take abrupt change detection into account.
>     1. **We compared DTF-net with both methods that consider and do not consider abrupt changes.** First, the $\ell_1$ and H-P methods consider abrupt changes using the sum of squared functions. Second, Wavelet is grounded in frequency, and EMD and Median rely on decomposition, which do not prioritize the consideration of abrupt changes. Finally, we also compared with CPD and anomaly detection algorithms, which consider abrupt changes.
>     2. We supplemented additional detailed explanations and results **in the revised version of Section 2.1 and Table 1** based on the comments.
> 3. Anomaly detection can be used instead of.
>     1. Thank you for your suggestion and **we added experiments on anomaly detection** methods [1] **in the revised version of Table 1.**
>     2. Additionally, we plan to add details and other various anomaly detection algorithms [1-3] in Appendix F, Discussion section. **If the methods that the reviewer wants to specify are not added, we want to hear from you through discussion.**
>
>     [1] Haixu Wu, Tengge Hu, Yong Liu, Hang Zhou, Jianmin Wang, and Mingsheng Long. Timesnet: Temporal 2d-variation modeling for general time series analysis. In International Conference on Learning Representations, 2023.
>
>     [2] *Yiyuan Yang, Chaoli Zhang, Tian Zhou, Qingsong Wen, and Liang Sun. 2023. DCdetector: Dual Attention Contrastive Representation Learning for Time Series Anomaly Detection. In Proceedings of the 29th ACM SIGKDD Conference on Knowledge Discovery and Data Mining (KDD '23). Association for Computing Machinery, New York, NY, USA, 3033–3045. https://doi.org/10.1145/3580305.3599295*
>
>     [3] Jiehui Xu, Haixu Wu, Jianmin Wang, & Mingsheng Long (2022). Anomaly Transformer: Time Series Anomaly Detection with Association Discrepancy. In *International Conference on Learning Representations*.
>
>
> We would again like to thank ****Reviewer Q1zH****, and we hope that our changes adequately address your concerns. Please let us know if you have any further questions or comments, and we are very happy to follow up!

---

> ### Author Response · Authors · 2023-11-21
> **Response to Reviewer Q1zH (2/)**
>
> We extend our gratitude to ****Reviewer Q1zH**** and present additional responses regarding anomaly detection experiments.
>
> 1. We introduced experiments on recent anomaly detection algorithms mentioned above [1-3] in Appendix F.3.
> 2. We identified two challenges for trend filtering in anomaly detection:
>     1. The sensitivity of the threshold to detect anomalies, particularly in capturing extreme value points.
>     2. Despite finding an optimal threshold for trend filtering, anomaly detection often misses peak points in complex sub-sequences.
>
> We would again like to thank ****Reviewer Q1zH****, and we hope that our changes adequately address your concerns. Please let us know if you have any further questions or comments, and we are very happy to follow up!

---

### Author Response · Authors · 2023-11-19
**General Response to All Reivewers (1/)**

We express our gratitude to all reviewers and in general response, we provide the updated experiment results.

### Table 1
|  | Linear Signal |  |  |  | Linear Signal + noise (0.2) |  |  |  |
| --- | --- | --- | --- | --- | --- | --- | --- | --- |
|  | 1) full-sequence |  | 2) abrupt-squence |  | 1) full-sequence |  | 2) abrupt-squence |  |
|  | MSE | MAE | MSE | MAE | MSE | MAE | MSE | MAE |
| ADAGA | 4.3507 | 1.4319 | 7.0179 | 1.8476 | 4.3434 | 1.4428 | 7.0120 | 1.8668 |
| RED-SDS | 0.9678 | 0.6566 | 1.6329 | 0.9169 | 1.0036 | 0.6782 | 1.6660 | 0.9365 |
| TimesNet | 3.0047 | 1.4112 | 3.2740 | 1.4161 | 3.0841 | 1.4204 | 3.3304 | 1.4364 |
| EMD  | 5.2836 | 1.7294 | 6.4599 | 1.8313 | 5.3096 | 1.7401 | 6.4410 | 1.8431 |
| Median  | 4.4506 | 1.5335 | 5.7018 | 1.8099 | 4.4766 | 1.5525 | 5.6859 | 1.8204 |
| H-P  | 0.1881 | 0.2807 | 0.2923 | 0.3493 | 0.2253 | 0.3311 | 0.3238 | 0.3934 |
| Wavelet  | 0.0427 | 0.1676 | 0.0451 | 0.1740 | $1e-30$ | $6e-16$ | $2e-30$ | $8e-16$ |
| $\ell_1$ ($\lambda$=0.1) | 0.0150 | 0.0885 | 0.0166 | 0.0971 | 0.0461 | 0.1703 | 0.0500 | 0.1807 |
| $\ell_1$ ($\lambda = 5e-4$) | 0.0379 | 0.1570 | 0.0403 | 0.1638 | 0.0004 | 0.0175 | 0.0004 | 0.0174 |
| DTF-net (ours) | 0.0378 | 0.1554 | 0.0389 | 0.1608 | 0.0289 | 0.0826 | 0.0286 | 0.0855 |

### Table 5
| State |  | non-sequential |  | non-sequential |  | sequential |  | sequential |  | zero padding |  |
| --- | --- | --- | --- | --- | --- | --- | --- | --- | --- | --- | --- |
|  |  | dynamic |  | static |  | dynamic |  | static |  |  |  |
| Metric |  | MSE | MAE | MSE | MAE | MSE | MAE | MSE | MAE | MSE | MAE |
| Exchange | 24 | 0.0250 | 0.1198 | 0.0263 | 0.1228 | 0.0264 | 0.1231 | 0.0263 | 0.1227 | 0.0259 | 0.1215 |
|  | 48 | 0.0487 | 0.1658 | 0.0500 | 0.1697 | 0.0496 | 0.1689 | 0.0501 | 0.1699 | 0.0486 | 0.1655 |
|  | 96 | 0.0983 | 0.2349 | 0.0995 | 0.2363 | 0.0994 | 0.2363 | 0.0982 | 0.2348 | 0.0983 | 0.2350 |
|  | 192 | 0.1983 | 0.3583 | 0.1986 | 0.3587 | 0.2013 | 0.3607 | 0.1984 | 0.3582 | 0.2003 | 0.3598 |
|  | 336 | 0.3160 | 0.4561 | 0.3163 | 0.4562 | 0.3166 | 0.4562 | 0.3140 | 0.4541 | 0.3166 | 0.4564 |
|  | 720 | 0.7933 | 0.6874 | 0.7922 | 0.6822 | 0.7903 | 0.6878 | 0.7893 | 0.6874 | 0.7889 | 0.6871 |
| ETTh1 | 24 | 0.0253 | 0.1205 | 0.0264 | 0.1228 | 0.0255 | 0.1205 | 0.0259 | 0.1214 | 0.0258 | 0.1216 |
|  | 48 | 0.0375 |  0.1479 | 0.0381 | 0.1487 | 0.0381 | 0.1485 | 0.0379 | 0.1478 | 0.0379 | 0.1479 |
|  | 96 | 0.0519 | 0.1740 | 0.0551 | 0.1810 | 0.0554 | 0.1808 | 0.0553 | 0.1808 | 0.0555 | 0.1811 |
|  | 192 | 0.0676 | 0.2013 | 0.0680 | 0.2019 | 0.0700 | 0.2051 | 0.0687 | 0.2026 | 0.0695 | 0.2041 |
|  | 336 | 0.0803 | 0.2247 | 0.0805 | 0.2252 | 0.0796 | 0.2238 | 0.0806 | 0.2254 | 0.0803 | 0.2244 |
|  | 720 | 0.0776 | 0.2224 | 0.0808 | 0.2271 | 0.0809 | 0.2273 | 0.0808 | 0.2273 | 0.0795 | 0.2255 |
| Illness | 24 | 0.5881 | 0.5358 | 0.5805 | 0.5363 | 0.5845 | 0.5376 | 0.5621 | 0.5316 | 0.5808 | 0.5464 |
|  | 48 | 0.6858 | 0.6359 | 0.6558 | 0.6329 | 0.6813 | 0.6535 | 0.6255 | 0.5964 | 0.6551 | 0.6310 |
|  | 60 | 0.6640 | 0.6423 | 0.7481 | 0.7029 | 0.6506 | 0.6265 | 0.7455 | 0.6979 | 0.6513 | 0.6270 |

We would again like to thank all reviewers, and we hope that our changes adequately address your concerns. Please let us know if you have any further questions or comments, and we are very happy to follow up!

---

### Author Response · Authors · 2023-11-21
**General Response to All Reivewers (2/)**

We extend our gratitude to all reviewers and present a concise overview of the modifications made in the revised version:

1. We included motivation for utilizing RL in Trend Point Detection.
2. A diagram of DTF-net has been incorporated in Figure 1.
3. Formulas in Section 3.1.2 were added to provide a comprehensive understanding of random sampling in segmentation and dynamic length.
4. The motivation for using the MSE cost function based on TSF as a reward was added in Section 3.2.
5. The reward process and its algorithm in Section 3.2 were supplemented.
6. An explanation of how reward sampling works and why it can mitigate overfitting issues was added in Section 3.2.
7. A clarification on why trend filtering analysis is challenging based on its ambiguity was added in Section 4.1.1.
8. Trend filtering analysis experiment results were supplemented in Figure 3 and Table 1.
9. A proof based on [1] in Appendix A was added for a more comprehensive understanding of the relation between approximation and the Gaussian distribution.
10. An ablation study on state random sampling on segmentation and dynamic length was added in Appendix C.1 (Table 5).
11. A discussion on how DTF-net addresses the limitation of expensive computation cost was supplemented in Appendix D.2.
12. A discussion on anomaly detection on trend filtering was added in Appendix F.3.
13. Multivariate time series forecasting was introduced as future work in Appendix G.

[1] Daizong Ding, Mi Zhang, Xudong Pan, Min Yang, and Xiangnan He. 2019. Modeling Extreme Events in Time Series Prediction. In Proceedings of the 25th ACM SIGKDD International Conference on Knowledge Discovery & Data Mining (KDD '19). Association for Computing Machinery, New York, NY, USA, 1114–1122. https://doi.org/10.1145/3292500.3330896

We would again like to thank all reviewers, and we hope that our changes adequately address your concerns. Please let us know if you have any further questions or comments, and we are very happy to follow up!

---

### Meta-Review · Area_Chair_2xSa · 2023-12-15

**Metareview:**

The paper presents the Dynamic Trend Filtering network (DTF-net), a new trend-filtering method for time series data that effectively identifies abrupt changes (Dynamic Trend Points) using a Markov Decision Process and Reinforcement Learning. However, the approach faces challenges in methodological clarity, comparison breadth with other techniques, and potential issues in training and computational requirements, suggesting areas for further development and research.

**Justification For Why Not Higher Score:**

The paper has several drawbacks as pointed out by the reviewers.

**Justification For Why Not Lower Score:**

NA.

---

### Decision · Program_Chairs · 2024-01-16

Reject